# A proximity-based *in silico* approach to identify redox-labile disulfide bonds: The example of FVIII

**Andrea Arsiccio**[1]◎, **Clive Metcalfe**[2]◎*, **Roberto Pisano**[1], **Sanj Raut**[2], **Carmen Coxon**[2]◎

**1** Department of Applied Science and Technology, Politecnico di Torino, Torino, Italy, **2** National Institute for Biological Standards and Control, Hertfordshire, United Kingdom

◎ These authors contributed equally to this work.
* clive.metcalfe@nibsc.org

**Data Availability Statement:** All relevant data are within the paper and its Supporting information files.

**Funding:** The authors received no specific funding for this work.

## Abstract

Allosteric disulfide bonds permit highly responsive, transient 'switch-like' properties that are ideal for processes like coagulation and inflammation that require rapid and localised responses to damage or injury. Haemophilia A (HA) is a rare bleeding disorder managed with exogenous coagulation factor(F) VIII products. FVIII has eight disulfide bonds and is known to be redox labile, but it is not known how reduction/oxidation affects the structure-function relationship, or its immunogenicity—a serious complication for 30% severe HA patients. Understanding how redox-mediated changes influence FVIII can inform molecular engineering strategies aimed at improving activity and stability, and reducing immunogenicity. FVIII is a challenging molecule to work with owing to its poor expression and instability so, in a proof-of-concept study, we used molecular dynamics (MD) to identify which disulfide bonds were most likely to be reduced and how this would affect structure/function; results were then experimentally verified. MD identified Cys1899-Cys1903 disulfide as the most likely to undergo reduction based on energy and proximity criteria. Further MD suggested this reduction led to a more open conformation. Here we present our findings and highlight the value of MD approaches.

## Introduction

Disulfide bonds are covalent bonds formed between the sulfur atoms of two cysteine residues in a protein. These bonds contribute to the mechanical stability of the protein tertiary structure and this stability can allow certain proteins to survive and function in harsh extracellular environments that can be rich in proteases and subject to large changes in salt and pH that would otherwise degrade a non-covalently stabilised protein structure [1]. However, it is becoming increasingly apparent that a subset of disulfide bonds are more susceptible to reduction than others and that these labile disulfide bonds can, when reduced, impart structural changes that translate to modulated function. These functional disulfide bonds are termed allosteric disulfide bonds and they permit rapid and reversible changes in protein structure that negates the need for energetically expensive and time consuming *de novo* synthesis [2] and are often referred to as "redox switches". There are many examples of functional allosteric disulfide

**Competing interests:** The authors have declared that no competing interests exist.

bonds in homeostatic processes including cell signalling [3], DNA repair [4], apoptosis [5], transcription and translation [6], coagulation [7, 8], and immunity [9–11], as well as pathophysiological conditions such as thrombosis [12, 13], infection [14, 15], inflammation [16], and cancer [17]. Processes such as those involved in blood clotting and host defence require rapid, transient, localised changes to protein structure and function; reduction of allosteric disulfide bonds is an ideal way to do this. The lability of allosteric disulfide bonds is exploited by thiol oxidoreductase enzymes that facilitate bond oxidation (making) and reduction (breaking) and therefore regulate the process. This large family of enzymes range from the small 14 kDa thioredoxin (Trx) and thioredoxin-like proteins [18, 19] to the larger, multi-domain protein disulfide isomerases (PDI) [20]; all mediate their function through their Trx domains.

Coagulation factors are proteins that work in concert with platelets to maintain haemostasis, forming clots at sites of damage to minimise blood loss. Both platelet and coagulation proteins are regulated by allosteric disulfide bonds [8, 12, 21–23] and there is increasing interest in developing small molecules that target thiol isomerases such as the PDIs to control platelet-induced thrombosis [12, 24, 25]. In addition to PDI and platelets, the role of Trx in regulating coagulation factors is also well established [13, 26]. More recently we demonstrated that Trx can affect platelet function and highlighted this enzyme as an important regulator of not just the plasma component of clotting but also the cellular component [27]. In light of the considerable efforts to develop small molecules to control platelet function through regulation of thiol isomerase activity, it is important to understand the extent to which they regulate coagulation and establish whether antiplatelet drugs designed to antagonise platelet function through oxidoreductase antagonism may negatively impact haemostasis through effects on the coagulation cascade. Furthermore, it is already known that some biotherapeutic drugs are sensitive to thiol isomerase activity [28] and it is therefore essential that we understand how allosteric disulfide bonds affect structure/function. Many coagulation factors and plasma proteins are important biotherapeutics for the treatment of rare bleeding disorders (e.g. factor VIII, factor IX, and bypassing agents for haemophilia) and understanding their susceptibility to redox-mediated effects will be important in maintaining their safety and efficacy. One such biotherapeutic is coagulation factor VIII (FVIII), a multidomain protein with the composition A1-A2-B-A3-C1-C2. The B-domain is highly glycosylated and dispensable for expression and function and many recombinant FVIII products lack the B-domain, as its loss improves protein yield without compromising activity. FVIII circulates as a heterodimer composed of the heavy chain (A1-A2) and the light chain (A3-C1-C2). This heterodimer contains eight disulfide bonds [29], seven of which are conserved in the highly homologous factor V (FV), which has the same architecture as FVIII, and ceruloplasmin, which shares the A1-A2-A3 arrangement. The only disulfide bond not conserved between the FVIII and FV structures is the exposed FVIII Cys1899-Cys1903 [29]. Work published over 40 years ago reported FVIII as a Trx substrate [26] and considering the efforts underway to develop oxidoreductase inhibitors it is important to understand how FVIII activity is affected by reduction. Not only may it have implications for the use of PDI inhibitors as antiplatelet drugs, but it may also help us to understand why conditions linked to increased Trx levels, such as inflammation and cancer, are also associated with changes in blood clotting. Key to understanding redox regulation of blood coagulation proteins and products is identifying key labile disulfide bonds that influence protein structure/function where dysregulation of these disulfide bonds in manufacture or *in vivo* after administration could compromise drug safety and efficacy. These labile bonds can be modified via molecular engineering or used in combination with oxidoreductase inhibitors, generating next generation biotherapeutics and/or improved treatment regimes. One of the biggest hurdles to engineering next generation proteins is identifying which disulfide bond(s) are labile. Both experimental and predictive modelling methods (discussed in [30]) have been

successful in identifying labile disulfide bonds with allosteric properties in many proteins, however these can be complex and time consuming processes.

In this study we describe a new predictive modelling method for identifying labile disulfide bonds within protein structures and validate it using known labile disulfide bonds reported in the literature. This approach is based on all-atom molecular dynamics (MD) simulations of proteins in explicit solvent and in the presence of tris(2-carboxyethyl)phosphine (TCEP), a reducing agent frequently used to break disulfide bonds. Our approach was then applied to predict which disulfide bonds are labile in coagulation FVIII. These were then experimentally validated and their effect on FVIII activity quantified.

## Materials and methods

### Molecular dynamics simulations for the identification of labile disulfide bonds

The interaction of TCEP with CD44 (PDB ID: 1UUH [31]), interleukin-2 receptor subunit gamma (CD132, PDB ID: 2ERJ [32]), interleukin 4 (IL-4, PDB ID: 3BPL [33]), tissue factor (TF, PDB ID: 1BOY [34]), the D1 and D2 domains of glycoprotein 130 (GP130, PDB ID: 1P9M [35]) and FVIII light chains [36] was investigated using molecular dynamics.

All the simulations described in the following were performed using Gromacs (5.1.4 or 2018.6) [37] and Plumed (2.4.1 or 2.5.1) [38], employing the GROMOS54A7 force field [39] for the protein and the SPC/E force field [40] for water. The topology file for TCEP was obtained from the automated topology builder (ATB) server [41], and a 100:1 TCEP to protein ratio was used in all simulation boxes.

For all the simulations described in the following we used periodic boundary conditions, and the cut-off radius for both Coulombic (calculated using the PME method [42]) and Lennard-Jones interactions was set to 1.2 nm. In all simulations, 1 native protein molecule was introduced into the simulation box, and the overall charge was neutralized using $Na^+$ or $Cl^-$ ions. Each box was then energy minimized using the steepest descent algorithm, and equilibrated for 1 ns in the NPT ensemble, using Berendsen pressure and temperature coupling [43]. The production runs were also performed in the NPT ensemble, controlling temperature with the Nose-Hoover thermostat (0.5 ps relaxation time) [44, 45], and pressure with the Parrinello-Rahman barostat (3 ps relaxation time) [46]. The LINCS algorithm was employed for constraining all bonds [47].

First, 3-replica multiple walker parallel bias metadynamics (PBMetaD [48]) was used to select a good proximity-based criterion for the identification of labile disulfides. Afterwards, multiple unbiased MD simulations exploiting this criterion were performed.

### Parallel bias metadynamics

Metadynamics [49] works by introducing a history-dependent bias potential $V(s_i,t)$ that acts on selected degrees of freedom of the system $s_i$, generally referred to as collective variables (CVs),

$$V(s_i, t) = \int_0^t \omega_i(t') \exp\left( -\frac{(s_i - s_i(t'))^2}{2\sigma_i^2} \right) dt'$$

where $\omega_i$ is a deposition hill height and $\sigma_i$ a Gaussian width.

Such bias potential pushes the simulated system out of local minima, promoting the exploration of a considerably larger fraction of the phase space compared to conventional MD simulations. The computational cost associated with the deposition of the bias, however, increases exponentially with the number of CVs selected. Parallel bias metadynamics alleviates this issue

by constructing multiple one-dimensional biases, each acting on a single CV in parallel. This makes it possible to include as many CVs as needed at a reasonable computational cost.

3-replica multiple walker PBMetaD simulations were performed at 310 K and 1 bar, for CD44 in presence of 100 TCEP molecules. The simulation box was cubic with 7.9 nm side length. Seventy nanoseconds per replica (210 ns total simulation time) were performed, and a 1 fs time step was used.

The protein radius of gyration ($R_g$), and the coordination number of the TCEP molecules (CN) around the protein were used as CVs. The Gaussian height was set to 2 kJ/mol, the bias factor to 15, and the Gaussian deposition rate to 1 hill/ps. The σ (Gaussian width) values used were 0.02 nm and 2 for $R_g$ and CN, respectively.

The radius of gyration for a molecular structure containing $N$ atoms is defined as,

$$R_g = \sqrt{\frac{\sum_i^N m_i (R_i - R_{COM})^2}{\sum_i^N m_i}} \tag{1}$$

where $R_{com}$ is the position of the centre of mass (COM), and $m_i$ and $R_i$ are the mass and position of atom $i$, respectively. This CV was used to improve the sampling of more expanded, but not unfolded, configurations of CD44. The bias factor and Gaussian height were in fact defined so as to impede complete unfolding.

The coordination number CN was defined as,

$$CN = \sum_{i=1}^{N_A} \left( \sum_{j=1}^{N_B} \frac{1 - \left(\frac{r_{ij}}{r_0}\right)^n}{1 - \left(\frac{r_{ij}}{r_0}\right)^m} \right) \tag{2}$$

where $r_{ij}$ is the distance between species $i$ in group A (containing $N_A$ entities) and $j$ in group $B$ (including $N_B$ entities), and $r_0$ is a reference distance. Group A was the protein COM, while group B comprised the centre of masses of all TCEP molecules. The value of $r_0$ was set to 3 nm, while the exponentials $n$ and $m$ were set to 6 and 12, respectively.

This preliminary 3-replica multiple walker PBMetaD simulation allowed us to select a proximity-based criterion for the identification of labile disulfide bonds. The selected parameter was the coordination number of TCEP molecules at 0.8 nm from the disulfide bond. This criterion was subsequently used in unbiased MD simulations.

## Unbiased MD simulations to identify labile disulfide bonds

The labile disulfides in some of the model proteins used have been experimentally validated (Cys77-Cys97 for CD44 [10], Cys160-Cys209 for CD132 [11], Cys3-Cys127 for IL-4 [11], and Cys186-Cys209 for tissue factor [7]). These proteins were preliminarily simulated in order to access and refine the accuracy of the simulation approach used. The same simulation approach was then extended to the light chains of FVIII. Snapshots of the model proteins used in this work, with highlighted disulfides, are shown in Fig 1.

For each of these proteins, we started at least 20 independent trajectories. Each protein was simulated in presence of 100 TCEP molecules, at 310 K and 1 bar, and using a 1 fs timestep. Simulation boxes were cubic, with side length equal to 10.7 nm for tissue factor, 10.2 nm for CD132, 7.9 nm for IL-4 and CD44, 10.6 nm for GP130 and 14.4 nm for FVIII. CD44 and CD132 were also simulated in presence of dithiothreitol (DTT), again at a 100:1 DTT:protein ratio. DTT is another common reducing agent, often used to break disulfide bonds in proteins. The topology file for DTT was also downloaded from the ATB server. The aggregated simulation time for these unbiased simulations was about 2.29 μs.

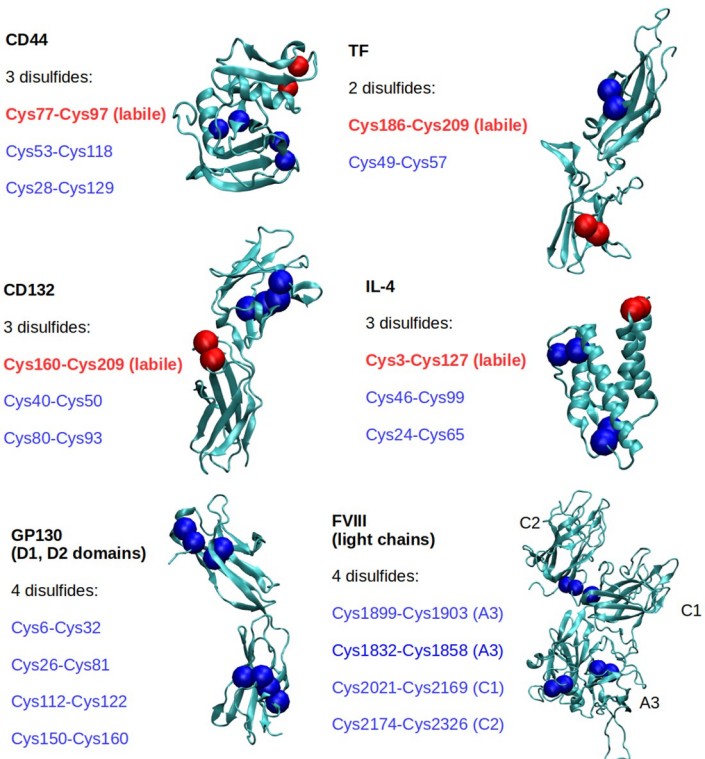

**Fig 1. Cartoon representation of the model proteins used for the molecular dynamics (MD) study.** The disulfide bonded cysteine residues are highlighted as coloured beads and listed on the left of each snapshot.

During these unbiased trajectories, we tracked the position of the TCEP (or DTT) molecules. We then applied two different criteria to identify a labile DSB [50], (1) distance-based or (2) distance+energy-based. According to the first criterion, a disulfide bond was considered to be reduced as soon as a TCEP molecule got closer than 0.8 nm. This same proximity-based condition had to be met for criterion number 2, but in this case a further requirement had to be satisfied. Specifically, the simulation was stopped, the disulfide bond was reduced and the resulting system energy minimized using the steepest descent algorithm. The initial structure, before disulfide bond reduction, was also energy minimized. The potential energies of the two systems, before ($E_{in}$) and after ($E_{fin}$) disulfide reduction, were compared and the disulfide deemed to be broken only if $E_{fin}<E_{in}$.

## Analysis of FVIII conformational changes upon reduction of the labile disulfide bond

After the identification of the FVIII labile disulfide bond in our trajectories, two additional simulations were performed for B-domain deleted FVIII (PDB ID: 3CDZ [36]) in water at 300 K and 1 bar, to investigate how reduction of the labile disulfide bond would affect protein conformation. In this case, both the heavy and the light chains of the protein were simulated.

The force fields used were the same previously described, but in this case no TCEP molecules were inserted into the simulation box. For one simulation, the labile disulfide bond was reduced before starting the trajectory, while native FVIII was used in the other case. The protein molecules were inserted into two separate cubic boxes with 14.9 nm side length. The

overall charge of the systems was neutralized by the addition of Cl$^-$ ions. The protein structures were simulated for 200 ns at 300 K and 1 bar in the NPT ensemble, using a 2 fs time step.

The protein radius of gyration ($R_g$), number of intra-molecular hydrogen bonds ($HB_{pp}$), RMSD (root mean square deviation) with respect to the initial structure and solvent accessible surface area (SASA) as function of simulation time were computed. The conformations assumed by FVIII were grouped together by performing a cluster analysis based on the Daura algorithm [51], after discarding the first 100 ns of the trajectory. More specifically, the conformations were grouped together if the root mean square deviations (RMSDs) of the protein backbone (N-Cα-C atoms) were less than 0.2 nm compared to each other. The most sampled protein conformation during the equilibrated trajectories (last 100 ns) was selected based on this analysis for the case of both the reduced and oxidized forms of FVIII. The change in secondary structure as a result of disulfide reduction was also investigated, by comparing the equilibrated structures obtained from the two simulations.

### Kinetic trapping of labile disulfide bonds in recombinant FVIII

FVIII products (diluted to 200 international units [IU] per ml) were incubated either with 1mM TCEP at room temperature (RT) for 15 minutes, or with Trx (1µM), NADPH (0.2 mM), and thioredoxin reductase (TrxR1, 1 nM) at 37˚C for 30 minutes. Alexa Fluor 488-maleimide was then added to label free thiols (30 min, room temperature, in the dark). Samples were then boiled in Lamelli buffer and separated on a 4–20% gel by SDS-PAGE and visualised using a GeneSys Imager. Total protein was evaluated using Coommassie blue.

### Quantitative mass spectrometry analysis of disulfide bonds in FVIII

50µg of recombinant FVIII was added to a number of 10kDa 500 µl centrifugal concentrators (Vivacon 500, Sartorious) and reduced with one of 1mM TCEP in PBS, 1mM DTT in PBS and Trx (1µM), NADPH (0.2 mM), TrxR1 (1 nM) for 90 min at 37˚C. After washing with 500 µl PBS, free cysteines arising from the reduction of disulfide bonds were alkylated with iodoacetamide (IAA) (100 µl of 50mM in 25mM ammonium bicarbonate for 1 h at room temperature in the dark). A non-reduced control sample was treated with IAA only to obtain the base level of free cysteines in FVIII. The samples were denatured and any remaining disulphide bonds TCEP-reduced (200 µl of 8 M urea solution with 25 mM TCEP for 1 h at room temperature) after which the tubes were centrifuged to remove liquid from filters. Cysteines arising from this were alkylated with NEM (200 µl of 50mM in PBS for 60 minutes at room temperature) to discriminate them from cysteines arising from the initial 1µm TCEP reduction of the intact FVIII. A 100% IAA labelled sample of FVIII was also prepared by denaturing and reducing 50µg in a centrifugal concentrator and alkylating with IAA as above. Three biological replicates were prepared and the samples were trypsin-digested as reported previously [52]. Briefly, samples were incubated with 2 µg trypsin (Sigma) in 100 µl Ambic at 37˚C overnight. Any remaining liquid was then spun into a clean collection tube, and 100 µl 0.1% formic acid was added to each sample for 10 min before centrifuging the liquid into the collection tube. Solutions of 50% and 100% acetonitrile with 0.1% formic acid were then consecutively added to the samples, which were centrifuged to remove liquid at each stage. The filters were then removed, and samples placed in a speedvac (Thermo) at 45˚C to remove all liquid. For mass spectrometry, samples were reconstituted in 0.1% formic acid, followed by sonication.

### Mass spectrometry

The tryptic peptide samples were injected as technical triplicates onto an Ultimate 3000 nano HPLC system coupled to an Orbitrap XL Discovery mass spectrometer (Thermo Scientific).

Samples were online desalted on a μ-Precolumn (C18 PepMap100, 300 μm id × 5 mm; 5 μm, 100 Å) at a flow rate of 25 μL/min, which was followed by separation on a nano analytical column (Acclaim PepMap100 C18, 75 μm id × 50 cm, 3 μm, 100 Å) (Thermo Scientific) using a 90-minute linear gradient from 5 to 40% solvent B (98% CH₃CN/2% H₂O/0.1% formic acid, v/v/v) versus solvent A (98% H₂O/2% CH₃CN/0.1% formic acid, v/v/v) at a flow rate of 300 nl/min. The mass spectrometer was operated in a data-dependent acquisition mode. The full survey scan (m/z 400–2000) was acquired in the Orbitrap with a resolution of 30,000 at m/z 400, which was followed by five MS/MS scans in which the most abundant peptide precursor ions detected in the preceding survey scan were dynamically selected and subjected for collision-induced dissociation (CID) in the LTQ (linear ion trap) to generate MS/MS spectra ('top-5 method').

## Quantitative analysis of disulfide bond REDOX state after reduction with TCEP

Data files of the mass spectrometry runs were combined and searched against the human Swiss-Prot database using Peaks 8 proteomics studio (Bioinformatics Solutions Inc. On, Canada). Precursor mass tolerance was 10 ppm and fragment ion tolerance was 0.6 Da with up to two missed trypsin cleavage sites per peptide allowed. Variable modifications were defined as deamidation on asparagine and glutamine, oxidation on methionine and alkylation with IAA or NEM (and hydrolysed variants) on cysteines. de-novo, peaks-db, SPIDER and peaks PTM algorithms were sequentially used to search against a concatenated target/decoy database, providing an empirical false discovery rate (FDR) and results are reported at a 1% target/decoy FDR for both peptides and proteins.

Peptides containing IAA labelled cysteines were selected for further analysis ensuring at least one cysteine from each disulfide bond was covered. Four control peptides from FVII, all of which contained no modifiable residues were selected and used for normalisation. All of the peptides used in the analysis are summarised in S1 Table. Precursor ion areas for the selected peptides were extracted using MS1 filtering in Skyline [53]. To normalise the data the ratio of precursor ion area of the peptide under analysis to the precursor ion areas of each of the control peptides is found for the control, Trx and 100% reduced samples. The % reduction for a given cysteine is calculated using the following equation for each peptide in each of the biological and technical replicates:

$$
\% \text{ reduction of unknown Cys peptide}
$$

$$
= \frac{\left[\left(\frac{\left(\frac{\text{Area of Cys peptide in unknown}}{\text{Area of control peptide 1 in unknown}}\right)}{\left(\frac{\text{Area of Cys peptide in 100\%}}{\text{Area of control peptide 1 in 100\%}}\right)} \times 100\right) + \left(\frac{\left(\frac{\text{Area of Cys peptide in unknown}}{\text{Area of control peptide 2 in unknown}}\right)}{\left(\frac{\text{Area of Cys peptide in 100\%}}{\text{Area of control peptide 2 in 100\%}}\right)} \times 100\right) + \ldots \left(\frac{\left(\frac{\text{Area of Cys peptide in unknown}}{\text{Area of control peptide } n \text{ in unknown}}\right)}{\left(\frac{\text{Area of Cys peptide in 100\%}}{\text{Area of control peptide } n \text{ in 100\%}}\right)} \times 100\right)\right]}{n}
$$

From this, the mean % reduction and standard deviation for each cysteine before and after reduction with DTT, TCEP and Trx was determined.

### FVIII activity assays

**Chromogenic assay.** FVIII activity was quantified using the Chromogenix Coatest® SP4 FVIII assay kit according to the manufacturers' instructions on an ACL Top 550 blood analyser (Werfen). In brief, FVIII was diluted to ~1 IU/ml in FVIII-deficient plasma and to this, activated FIXa, FX, $Ca^{2+}$, phospholipid, and the FXa substrate S-2765 were added to the appropriate concentrations and conversion of the substrate to a coloured product was measured at 405 nm.

### Enzyme-linked immunosorbent assay (ELISA)

Nunc Maxisorb plates were coated with 500 ng recombinant B domain deleted (BDD) FVIII for 2 hours at room temperature. Wells were then incubated with either PBS or 1 mM TCEP for 30 minutes at room temperature before being blocked with PBS/1% bovine serum albumin (BSA)/0.05% Tween 20 for 30 minutes (at room temperature). Recombinant anti-FVIII antibodies were added to the wells (100 μl, diluted in blocking buffer) and incubated for 1 hour at room temperature with gentle agitation. Unbound antibody was removed by washing (3 x 200 μl PBS/0.2% Tween 20) and anti-FVIII antibodies detected using an anti-human k chain antibody (Sigma Aldrich #A7164) conjugated to horse radish peroxidase (1:20,000) and 3,3′,5,5′-Tetramethylbenzidine (TMB) substrate. Plates were read on a microplate reader (SpectraMax M2e, Molecular Devices) as an end point assay.

### Statistics

Chromogenic assay and one-stage clotting assay (OSCA) data were analysed using a parallel line model (with a log transformation for the OSCA) and evaluated for linearity and parallelism using Combistats Version60. Data represent three experimental replicates.

## Results

### Haemostasis-related plasma proteins are substrates of Trx and contain labile disulfide bonds

A major driving force for this work is to understand the redox lability of coagulation factor (F) VIII. The redox lability of several haemostasis-related plasma proteins has been experimentally validated and we began by using these to validate our reduction protocols. Samples were incubated under chemical or enzymic reducing conditions before free cysteines were labelled with a thiol-reactive maleimide probe conjugated to a fluorescent dye. Samples were then separated by SDS-PAGE and visualised. As shown in the upper panel of Fig 2A, both enzymic and chemical reduction results in the appearance of bands corresponding to the proteins of interest, or an increase in their intensity, indicative of an increase in free cysteines; this confirmed we were able to reduce labile disulfides using both TCEP and thioredoxin and detect cysteine labelling in proteins previously reported to contain labile disulfide bonds, including factor (F) XI, von Willebrand factor (VWF), and beta glycoprotein I [13, 21, 54–57]. Furthermore, in agreement with previous reports [26], we can also detect an increase in free cysteine labelling when recombinant B domain deleted (BDD) FVIII (BDD rFVIII) is incubated under chemical or enzymic reducing conditions (Fig 2B). The light chain exhibits more intense staining than the heavy chain, which is noteworthy because the light chain conveys the majority of FVIII functionality (FIXa binding, VWF binding, and the interaction with phospholipids). We next set out to develop an *in silico* method to identify *which* of the 4 disulfide bonds in the FVIII light chain are the most likely to be redox labile using a molecular dynamics (MD) approach.

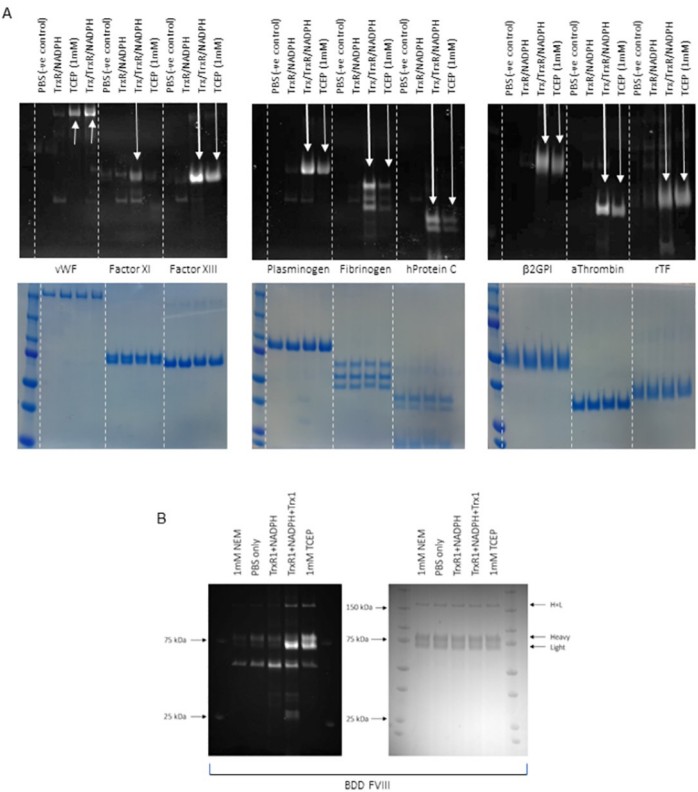

**Fig 2. TCEP and Trx reduce plasma proteins important for coagulation and haemostasis.** (A) Protein samples were reduced with thioredoxin or TCEP and free thiols labelled with Alexa 488-maleimide. Samples were separated on a 4–20% SDS-PAGE gel and visualised for the incorporation of the fluorescent label (top). Coomassie stains to show equal loading is also shown (bottom). (B) FVIII undergoes reduction by both thioredoxin and TCEP (Alexa 488-maleimide staining on the left, total protein [Coomassie staining] on the right).

## Identification of suitable proximity criteria to predict disulfide bond redox susceptibility

Using a simple proximity criterion previously used to investigate thiol-disulfide isomerization in a mutated immunoglobulin [50], we looked at the interaction of TCEP (tris(2-carboxyethyl) phosphine) with proteins with experimentally validated redox-labile disulfide bonds (Fig 1) before applying the method to the FVIII light chain. We first applied the parallel bias metadynamics method, which allows an extensive sampling of configuration space, to select a good proximity-based criterion for the identification of labile disulfides in proteins. We started with the Cys77-Cys97 disulfide bond in CD44 looking at three parameters– 1) the average distance between TCEP and the disulfide bonds, 2) the minimum distance between TCEP and the disulfide bonds, and 3) the average number of TCEP molecules whose centre of mass (COM) was closer than 0.8 nm to the COM of the disulfides. This 0.8 nm cut-off was selected as it is both small enough to capture local phenomena and large enough to accommodate at least a few TCEP molecules.

The average distance between TCEP molecules and disulfide bonds (parameter 1) is lowest for Cys53-Cys118 (Fig 3A), but this disulfide bond is not reduced under experimental conditions indicating that this parameter is not suitable for predicting labile disulfide bonds. The reason for this becomes evident when considering the data in Fig 3B (parameter 2), which

reveals that although TCEP can get close to the disulfide bond, TCEP molecules cannot get sufficiently close to react with it. The average distance parameter alone (Fig 3A) is not a reliable predictor for disulfide bond lability, and although the minimum distance parameter (Fig 3B) can identify labile disulfides in the framework of a proximity criterion, it is not, at least in this case, sensitive enough to distinguish between Cys77-Cys97, and Cys28-Cys129. The third criterion—the number of TCEP molecules that, on average, were closer than 0.8 nm to the COM of the disulfide bonds (Fig 3C)—allowed clearer separation of disulfide bonds with increased sensitivity compared with the minimum distance parameter. In our MD simulations, the CD44 disulfide Cys77-Cys97 is surrounded by the highest number of TCEP molecules (a proxy for concentration) when a 0.8 nm sphere around the COM is defined. This result is in line with experimental data for CD44 that identified Cys77-Cys97 as redox-labile [10].

However, the reduction of a disulfide bond is the result of both accessibility (i.e., TCEP molecules need to get close enough to react) and net energy gain (i.e., the reduced system should be energetically favoured). A set of unbiased simulations was therefore started for each protein shown in Fig 1, using either a (1) distance- or (2) a distance+energy-based criterion, as detailed in the Materials and Methods section. According to the distance-based criterion, the disulfide bond was considered to be reduced as soon as a TCEP molecule got closer than 0.8 nm. In contrast, a second requirement had to be satisfied according to criterion (2)—the potential energy of the reduced ($E_{fin}$) system had to be smaller than that of the oxidized ($E_{in}$) one ($E_{fin} < E_{in}$). The results of these unbiased simulations are shown in Fig 4. The average solvent accessible surface area (SASA) and root mean square deviation (RMSD) of each disulfide-forming cysteine during the unbiased simulations are displayed in S1 and S2 Figs.

The two criteria (distance or distance+energy) were applied to 5 experimentally validated redox-labile disulfide bonds including CD44 (Fig 4A), TF (Fig 4B), CD132 (Fig 4C), IL-4 (Fig 4D), and the D1 and D2 domains of GP130 (Fig 4E). The distance-only criterion (plain black bars) was enough to identify unambiguously the labile disulfides Cys77-Cys97 in CD44 (Fig 4A), Cys186-Cys209 in TF (Fig 4B), Cys160-Cys209 in CD132 (Fig 4C), Cys3-Cys127 in IL4 (Fig 4D), and Cys6-Cys32 in GP130 (Fig 4E). In this case, adding the energy-based requirement (dashed bars) did not modify the main conclusions. For instance, the distance-only criterion predicted that Cys77-Cys97 in CD44 would have 96.7% probability to be reduced, and this probability further increased to 100% when the distance+energy-based criterion was applied. The average SASA of the different disulfide bonds correlates fairly well with their lability (see S1 Fig) and is therefore a good initial approximation. However, it is important to note that the distance-based criterion is not only a measure of solvent exposure, but also takes into account the specific interactions between the probe (TCEP) and the protein surface. For instance, in our simulations of CD44 with TCEP, the solvent accessibility of Cys77-Cys97 was similar to that of Cys28-Cys129, or even lower, as shown in S1A Fig. However, TCEP molecules mostly remained closer to Cys77-Cys97, presumably because of specific interactions with this patch on the protein surface. Moreover, it is important to note that the solvent accessibility, as measured in MD simulations, is representative of protein dynamics, as such being more reliable than the solvent accessibility measured from static protein structures.

We also studied the interactions between dithiothreitol (DTT) and CD44 or CD132 (S3 Fig). DTT is smaller than TCEP (the radius of gyration of DTT is ≈0.26 nm, while it is ≈0.37 nm for TCEP). For this reason, a smaller cut-off (0.6 nm, instead of 0.8 nm) was used to estimate proximity of the DTT molecules. We found that the distance-based criterion could not distinguish between Cys77-Cys97 and Cys28-Cys129 in CD44 (S3A Fig), while adding the energy-based requirement succeeded in doing so, identifying Cys77-Cys97 as redox labile with 60% probability. The difference between S3A Fig (CD44-DTT interaction) and 4A (CD44-TCEP interaction) indicates that the selected probe, DTT or TCEP, plays a role (as

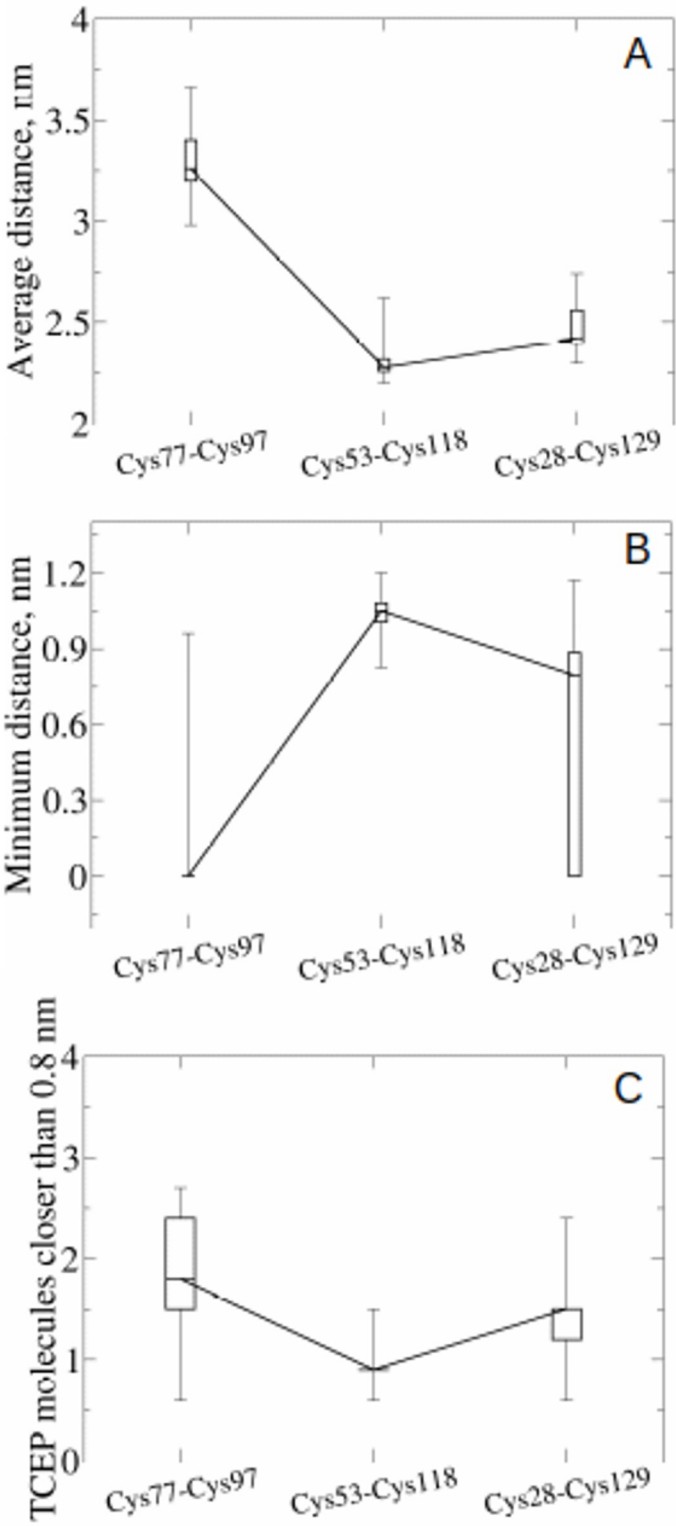

**Fig 3. Selecting proximity criteria from parallel bias metadynamics.** Boxplots showing (A) the average distance, (B) the minimum distance, or (C) the number of TCEP molecules closer than 0.8 nm to the disulfide bonds of CD44. In the boxplots, the top bar is the maximum observation, the lower bar is the minimum observation, the top of the box is the third quartile, the bottom of the box is the first quartile, while the middle bar represents the median. The medians are connected by a straight line to guide the eye.

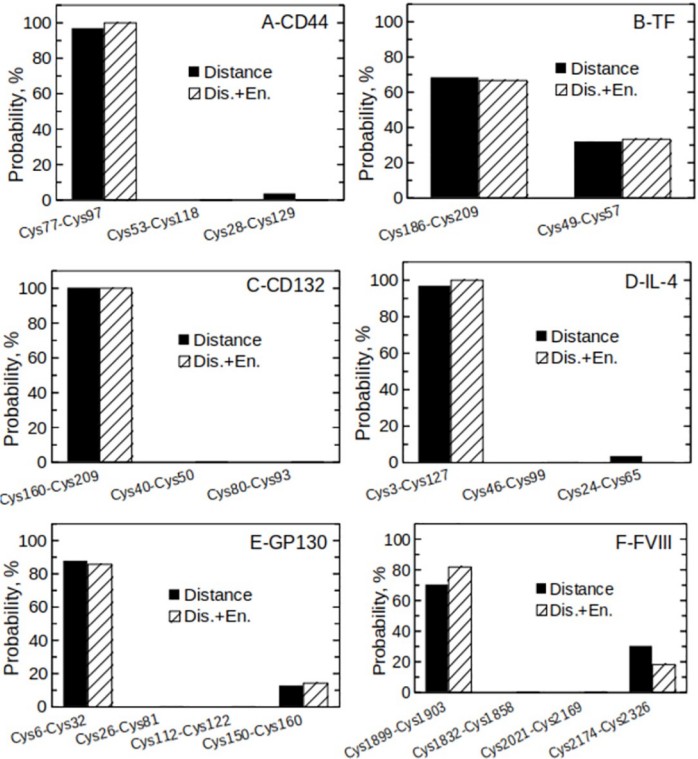

**Fig 4. Distance- or distance+energy-based criteria to identify redox-labile disulfide bonds.** Histograms showing the probability that each disulfide is reduced by TCEP according to the distance- (plain black bars) or distance+energy-based (dashed bars) criteria. The different panels refer to CD44 (A), tissue factor (B), CD132 (C), IL-4 (D), the D1 and D2 domains of GP130 (E), and FVIII light chains (F).

different probes preferentially interact with different patches on the protein surface), although the ultimate conclusions on redox lability are not affected, when the distance+energy-based criterion is used. S3B Fig further shows that both the distance and distance+energy-based criteria succeeded in identifying Cys160-Cys209 in CD132 as redox labile also in the case of DTT as molecular probe.

We next applied our *in silico* method to FVIII. The relatively large size of FVIII presents a challenge for *in silico* modelling with regards to computational costs, so we ran simulations for the light chains only (Fig 4F) in this instance. FVIII light chains convey most of the functionality and we had already identified this part of the molecule as being of interest due to the considerable increase in free thiols following thioredoxin or TCEP treatment (Fig 2B). Modelling of the FVIII light chains reveals that a larger number of TCEP molecules can get close to Cys1899-Cys1903 than to Cys1832-Cys1858, Cys2021-Cys2169 or Cys2174-Cys2326, therefore suggesting that Cys1899-Cys1903 is the most likely to be redox labile (70.0% probability according to the distance-based criterion, plain black bars). Applying the energy-based condition further augments the selectivity, indicating Cys1899-Cys1903 as redox-labile with an 81.8% degree of confidence. Finally, it is also interesting to note, looking at S2 Fig, that there is no evident correlation between the lability of a disulfide bond and its RMSD during the MD trajectories.

**Table 1. Structural parameters of disulfide bonds using online prediction analysis.** Four published FVIII structures were evaluated in parallel using the UNSW DSB prediction tool [58]. Allosteric disulfide bonds tend to be solvent accessible and have short alpha carbon distances. Stable structural disulphide bonds tend to have a -LHspiral conformation; these have the most common DSB conformation and have the lowest dihedral strain energy.

| | | Solvent Accessibility Å2 | | | | | Cα-Cα distance | | | | | 3D conformation | | | |
|---|---|---|---|---|---|---|---|---|---|---|---|---|---|---|---|
| | | 3CDZ | 6MF2 | 2R7E | 4BDV | Ave | 3CDZ | 6MF2 | 2R7E | 4BDV | Ave | 3CDZ | 6MF2 | 2R7E | 4BDV |
| HEAVY | Cys153-Cys179 | 7 | 1 | 64 | 16 | 22 | 4.54 | 3.93 | 4.44 | 4.35 | 4.32 | -/+LHHook | -LHStaple | -RHStaple | -RHStaple |
| | Cys248-Cys329 | 68 | 49 | 38 | 57 | 53 | 5.90 | 5.23 | 5.58 | 5.35 | 5.52 | -LHSpiral | -LHHook | -LHHook | -LHSpiral |
| | Cys528-Cys554 | 10 | 4 | 6 | 12 | 8 | 4.22 | 4.00 | 4.06 | 3.90 | 4.04 | -RHStaple | -RHStaple | -RHStaple | -RHStaple |
| | Cys630-Cys711 | 84 | 158 | 72 | 72 | 96.5 | 6.21 | 5.91 | 4.29 | 5.82 | 5.55 | +/-LHHook | +/-RHStaple | -LHHook | +/-LHHook |
| LIGHT | Cys1832-Cys1858 | 2 | 0 | 0 | 3 | 1.25 | 4.49 | 3.86 | 3.91 | 4.01 | 4.07 | -LHStaple | -LHHook | -RHStaple | -RHStaple |
| | **Cys1899-Cys1903** | 52 | | | | 52 | 4.97 | | | | 4.97 | -RHHook | | | |
| | Cys2021-Cys2169 | 6 | 9 | 50 | 7 | 18 | 6.67 | 6.11 | 6.16 | 6.23 | 6.29 | +/-RHHook | -LHSpiral | +/-RHHook | +LHStaple |
| | Cys2174-Cys2326 | 36 | 30 | 20 | 22 | 27 | 5.93 | 5.12 | 5.38 | 5.71 | 5.53 | +LHSpiral | +/-RHStaple | -/+LHHook | +/-RHStaple |

## Experimental validation of *in silico* predicted redox-labile Cys1899-Cys1903

Where suitable structures are available, some prediction can be made with regards to whether a disulfide bond may be redox labile or not. Using structural parameters like solvent accessibility or the distance between the alpha-carbons of the constituent cysteines and their 3D conformation [56], one can get an indication of how redox labile a given disulfide bond may be. Allosteric disulfide bonds tend to have high torsional bond strain as depicted by shorter alpha carbon distances; the −RHStaple and the −/+RHHook atomic arrangements are particularly prevalent conformations for allosteric disulfide bonds and more likely to undergo cleavage [57]. Using an online tool [58] we assessed redox labile disulfide bonds in published FVIII structures and their structural parameters are shown in Table 1. There is a degree of variability between FVIII structures due to the facts that crystallographic structures are snapshots of a flexible molecule—proteins are dynamic structures that have intrinsic flexibility to permits movement. The light chain disulfide bond in the A3 domain (Cys1899-Cys1903) predicted to be labile by our MD simulation is absent from three of the four structures and shown as reduced. As labile disulfide bonds are often broken by the reducing environment in the x-ray beam during data collection the lack of this disulfide bonds in the three structures lends some support to the MD prediction. Although only one set of parameters are available for this bond, when these are compared to the other bonds (average parameters over the four structures) it is the only one with high ($>30$ A$^2$) solvent accessibility *and* a short ($<5$ Å) distance between its alpha carbons. The other bonds are either poorly solvent exposed *or* have a long alpha carbon distance.

We applied a quantitative mass spectrometry and kinetic trapping of cysteine redox state method previously used to quantitate labile disulfide bonds in monoclonal antibodies [50] to examine the redox state of each disulfide bond in FVIII treated with chemical or enzymic reducing agents (Fig 5). The redox state of the disulfide bonds was determined from peptides containing at least one cysteine from each disulfide bond (details shown in S1 Table) except for the Cys630-Cys711 bond as there were no suitable tryptic peptides to monitor (the cysteine numbering from the mass spectrometry includes the 19 amino acid signal peptide so the numbers are 19 higher than the numbering in the parentheses which is consistent with the rest of the data). None of the disulfide bonds showed reduction in the control FVIII with the exception of Cys1899-Cys1903, which was approximately 20% reduced. However, following DTT, TCEP and Trx treatment, the Cys1899-Cys1903 disulfide bond was further reduced to ~100%

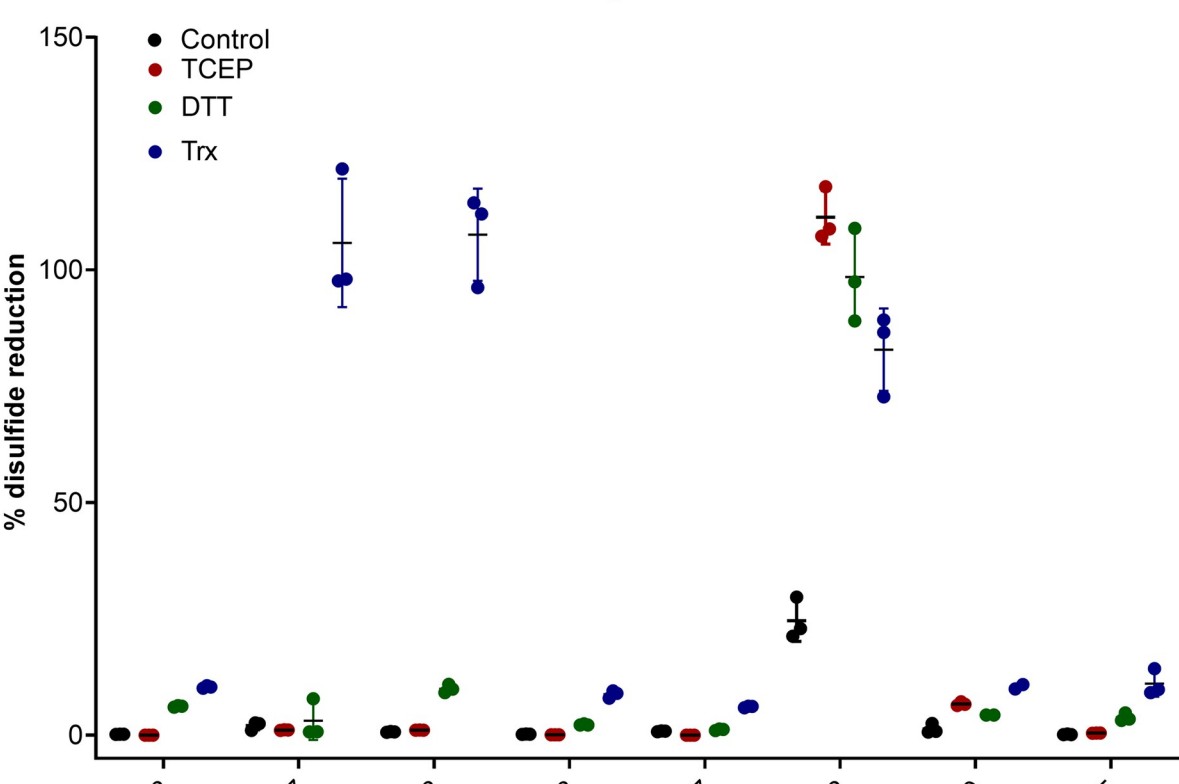

**Fig 5. Experimental validation of *in silico* MD simulations by quantitative mass spectroscopy and ELISA.** (A) Native or reduced (TRX 1) BDD rFVIII was alkylated with iodoacetamide (IAA) and subjected to tryptic digest and LC-MS/MS. Peak areas of IAA labelled cysteine peptides were compared to a control sample of BDD rFVIII which had been fully denatured, reduced and alkylated. The x-axis labels are as follows: cysteine 198 (from the Cys153-Cys179 bond); cysteines 267 and 348 (from the Cys248-Cys329 bond); cysteine 573 (from the Cys528-Cys554 bond); cysteine 1877 (from the Cys1832-Cys1858 bond); cysteine 1922 (from the Cys1899-Cys1903 bond); cysteine 2040 (from the Cys2021-Cys2169 bond) and cysteine 2345 (from the Cys2174-Cys2326 bond).

indicating it is labile and supporting the results obtained from the MD simulations. Interestingly, Trx treatment also led to the reduction of Cys248-Cys329, which sits in the heavy chain of FVIII. This disulfide bond was also fully reduced revealing that it is labile towards enzymatic Trx reduction but not chemical (DTT or TCEP) reduction.

### Effect of Cys1899-Cys1903 reduction on FVIII structure

The data shown in Fig 5 supports the FVIII light chain *in silico* modelling that predicted Cys1899-Cys1903 is redox labile. We next returned to *in silico* approaches to try and shed light on how reduction of Cys1899-Cys1903 disulfide bond could affect FVIII structure and function, this time running simulations for the heavy *and* light chains of FVIII in its oxidized and reduced (broken Cys1899-Cys1903 bond) forms. It is important to note that enhanced sampling methods were impracticable for these simulations due to the very large size of FVIII, and we therefore conducted conventional MD runs. We extended the simulation time until the

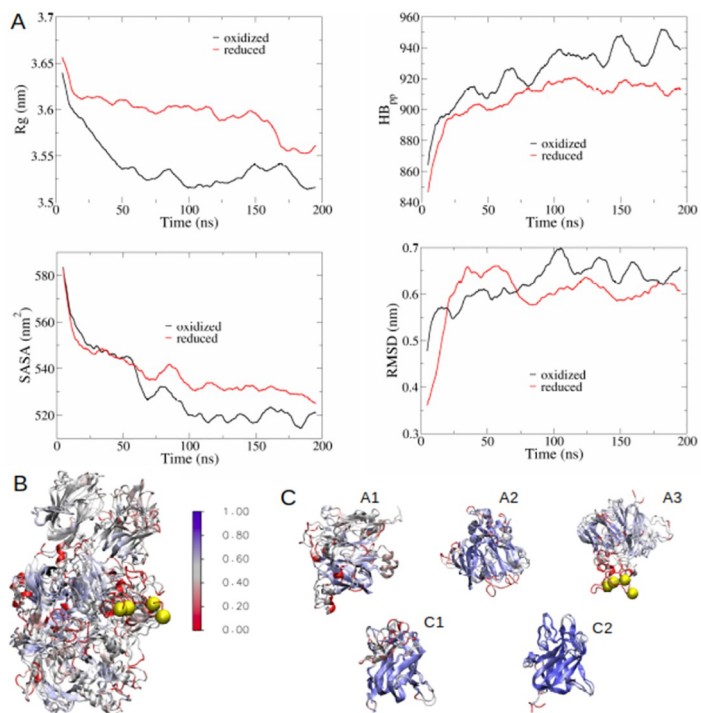

**Fig 6. Modelled conformational changes in FVIII upon reduction of disulfide bond Cys1899-Cys1903.** (A) Evolution of FVIII radius of gyration $R_g$, number of intra-molecular hydrogen bonds $HB_{pp}$, solvent accessible surface area SASA, and backbone RMSD with respect to the crystal structure, as function of simulation time, for both full-oxidized FVIII (black curve) and reduced FVIII (red curve). (B-C) Cartoon representation and superimposition of full-oxidized and reduced (Cys1899-Cys1903) FVIII. The alignment has been performed either on the whole structure (B), or on the separate domains (C). The cartoons are coloured according to the Q parameter, which is 1 for identical structures; the blue areas indicate structural similarity between full-oxidized and reduced FVIII, where Q = 1. Areas of poor structural similarity (Q = 0.1–0.3) are shown in red. The figure was obtained using the VMD software [59] and the STAMP algorithm [60], that works by minimizing the $C_\alpha$ distance between aligned residues of each molecule. The Cys1899-Cys1903 disulfide bond is represented as yellow beads.

computed properties (radius of gyration, internal hydrogen bonding network, solvent accessible surface area and backbone RMSD compared to the crystal structure, as shown in Fig 6) converged to stable values. The results of the conventional MD simulations presented are therefore a good indication of what may happen immediately after reduction of the labile disulfide bond; however, because of the impossibility to use enhanced sampling techniques, the reader should be aware that they may be not completely representative of long-time shifts in the protein conformation.

The radius of gyration $R_g$ decreased for both the oxidized and reduced forms compared to the crystal (static) structure (Fig 6A), with the reduced molecule having a slightly more expanded structure ($R_g \approx 3.58$ nm compared to $R_g \approx 3.53$ nm for the oxidized conformation) by the end of the simulation. This corresponded with a modest decrease in intra-molecular hydrogen bonds $HB_{pp}$ ($\approx 938$ for the oxidized conformation and $\approx 915$ when the disulfide is reduced) and an increase in the solvent-accessible surface area (SASA) ($\approx 519$ nm$^2$ for the oxidized protein compared to $\approx 531$ nm$^2$ with the reduced molecule).

The most sampled protein conformations (according to the Daura algorithm{Daura, 1999 #809}) for the equilibrated trajectories (last 100 ns) are shown in Fig 6B. Fig 6C shows an alignment of the different domains of FVIII in the oxidized and reduced forms. The protein regions

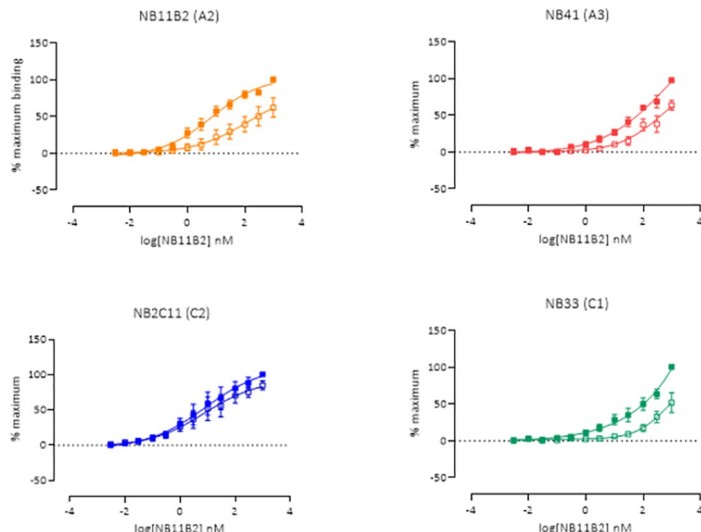

**Fig 7. ELISA analysis of FVIII-mAb binding in solid state.** Reduction of rFL-FVIII decreases binding of the A2- and A3-specific anti-FVIII antibodies NB11B2 and NB41. The binding of the C1 domain-specific antibody NB33 also decreases, while no change in binding is observed for the C2-specific antibody NB2C11.

subject to the largest conformational change upon reduction of the disulfide seemed to be residues 200–230 (A1 domain) and 555–570 (A2 domain) in the case of the heavy chain, and residues 1706–1732 (A3 domain), 1790–1804 (A3 domain) and 1880–1920 (A3 domain, this last region includes the reduced disulfide) in the light chain. Some slight conformational changes were also noted for the C1 domain (Fig 6C), while the C2 domain remained mostly unaltered. Using a panel of patient-derived FVIII neutralising antibodies developed in-house that bind to different regions of the FVIII molecule, we next evaluated FVIII-mAb binding in solid state using an ELISA. Fig 7 shows that, in support of the *in silico* modelling, reduction of recombinant full-length human FVIII (rFL-FVIII) decreases binding of the A2- and A3-specific anti-FVIII antibodies (NB11B2 and NB41, respectively). We also observe a decrease in the binding of the C1 domain-specific antibody NB33. No change in binding of the C2-specific antibody NB2C11 is observed. These data suggest that reduction of the Cys1899-Cys1903 disulfide bond results in a change in the conformation of FVIII through the A2, A3, and C1 domains.

## Effect of DSB reduction on FVIII activity

We next evaluated the effect of reduction on the activity of a number of FVIII products, including recombinant B domain-deleted (BDD) porcine FVIII (rpFVIII), three different recombinant full-length human FVIII (rFL-FVIII), recombinant human (rBDD FVIII), and a human plasma-derived FVIII product (pdFVIII), which also contains von Willebrand factor (VWF). In agreement with the mass spectroscopy data, FVIII is redox labile with an increase in free thiols upon reduction (Fig 8). The effect of reduction on FVIII activity was next assessed using the chromogenic assay, which is used clinically to measure FVIII activity in patient plasma. FVIII is not an enzyme so its activity (function) is measured indirectly by quantifying the activity of the tenase complex (factor IXa and factor Xa) whose formation it catalyses. The chromogenic assay is comprised of separate kit components (factor IXa, factor X, $Ca^{2+}$, phospholipids, and a tenase substrate that produces a coloured product) to which FVIII is added—product formation is proportional to the amount of 'active' (functional) FVIII in the sample.

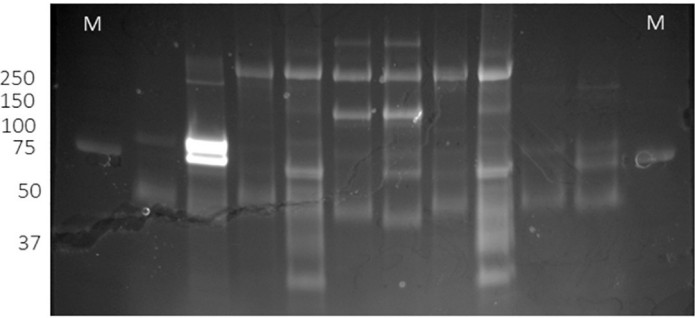

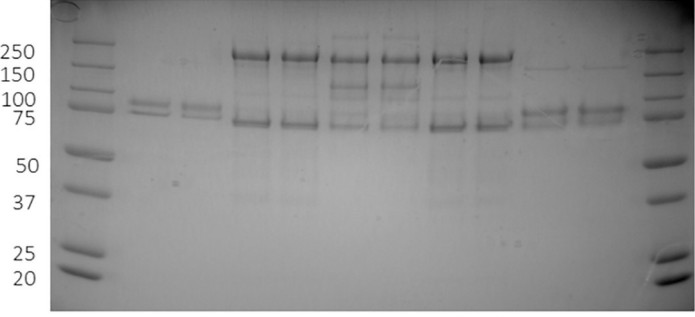

**Fig 8. TCEP-mediated reduction of a selection of FVIII products.** A selection of FVIII products were incubated with PBS or reduced with 1 mM TCEP and free -SH groups visualised with the Alexa Fluor 488-maleimide fluorescent label. FVIII products tested include recombinant porcine FVIII (rpFVIII) produced in Baby Hamster Kidney (BHK) cells, two recombinant full-length FVIII (rFL FVIII) products also produced in BHK cells, a rFL-FVIII product produced in Chinese Hamster Ovary (CHO) cells, and a recombinant B domain deleted FVIII product produced in BHK cells.

As shown in Fig 9, not all products behaved the same way when reduced. Recombinant porcine (BDD, Fig 9A) had marginally lower activity when reduced with DTT or TCEP, but reduction with Trx inhibited FVIII activity by ~80%. In contrast, recombinant *human* BDD FVIII showed an *increase* in activity Fig 9B. We looked at three full length (FL) recombinant rFVIII products, two expressed in BHK call (Fig 9C and 9D) and one manufactured in CHO cells (E) and found that, again, there were differences in how they behaved upon reduction, with two products showing between 50–80% decrease in activity with Trx but only modest decreases in activity when exposed to chemical reducing agents (C and E). The increased effects seen with Trx over the chemical reducing agents can be attributed to the additional reduction of Cys248-Cys329 in the heavy chain (Fig 5). However, production in different cells lines gives rise to proteins with different glycosylation patterns and other post-translational modifications, and additional differences in production/manufacturing, processing, and excipients could all have an effect on the state of the final product, but it is not known how the data shown here relate to these processes. The plasma derived product was inhibited by reducing agents but there was little difference between enzymic and chemical (Fig 9F). In summary, Trx has the capacity to reduce FVIII disulfides but the effect of these changes on FVIII activity varies from product to product (summarised in Fig 9G).

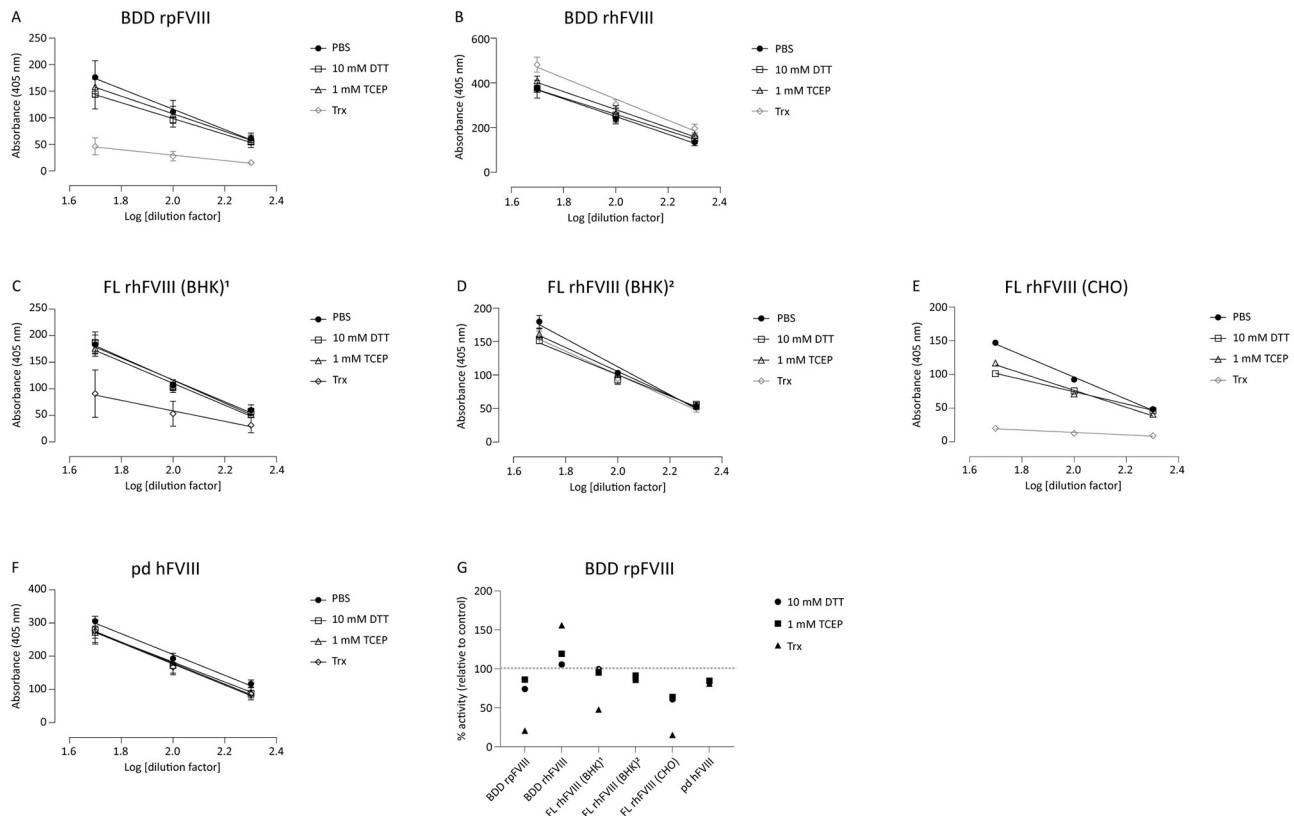

**Fig 9. Chromogenic activity assay of native and reduced (1mM TCEP) FVIII products.** Different FVIII products (porcine and human, FL and recombinant, CHO and BHK expressed) were incubated with either PBS (black circles), 10 mM DTT (squares), 1 mM TCEP (triangles), or 10 nM Trx (grey diamonds) before being assessed for activity (A–F). Activity was determined by parallel line analysis (G).

## Discussion

Molecular dynamics is an extremely powerful tool for studying protein structure/function, especially for proteins that are difficult to work with due to their inherent instability. Molecular dynamics (MD) [59, 60] can also provide a detailed picture of events often not directly measurable by existing experimental approaches. In the example here, we used MD to inform and guide experimental approaches to understand how reduction/oxidation affects the activity of coagulation factor(F) VIII. FVIII is an important biotherapeutic used to manage bleeding in haemophilia A (HA) patients but it is a labile protein and known to induce the formation of anti-drug antibodies in approximately one third of severe HA patients. It is known that FVIII is redox labile but there is some ambiguity about how reduction/oxidation affects this essential clotting factor (see below); identifying labile and allosteric disulfide bonds can help understand structure-function relationships and provide a basis for rational, hypothesis-driven protein engineering to improve stability and activity and reduce drug immunogenicity. We therefore applied a molecular dynamics approach to expand upon previously reported observations, resolve contradictory findings, and develop a hypothesis relating to how reduction of FVIII leads to changes in its activity. We applied a proximity criterion that correlates labile disulfide bonds with their accessibility, adding to previous studies that showed accessibility is a key factor [50]. To identify the proximity-based criterion, we made use of metadynamics [49], an enhanced sampling method that dramatically reduces the amount of computational time

required to obtain a significant sampling of configuration space. We also showed the importance of adding an energy-based criterion for the correct identification of labile disulfides. This simple computational approach permits simulations for whole proteins at a detailed, atomistic level, with a reasonable computational cost. We then used classical MD simulations to develop hypotheses relating to how the change in protein conformation that results from reduction of the identified labile disulfide bond affects FVIII activity.

We found the concentration of TCEP at a distance of <0.8 nm from the centre of mass (COM) of the disulfide bond was a good predictor of disulfide bond lability, especially when combined with an energy-based condition, requiring that the disulfide reduction was energetically favourable. This distance+energy-based condition was able to successfully identify experimentally validated allosteric disulfide bonds in a number of proteins. Applying this parameter to the FVIII light chains indicated Cys1899-Cys1903 disulfide bond was the most likely to undergo reduction. This particular disulfide bond has been the subject of previous studies and it is also noteworthy that it is the only disulfide bond that is unique to FVIII compared to FV. Disruption of this bond (by mutation of Cys1903 to either glycine or serine) was reported to improve expression of recombinant B domain deleted (BDD) FVIII by 2-fold, with no impact on activity [61, 62], which is a little surprising considering it resides in a part of the FVIII molecule that mediates FIXa binding [63]. Indeed, the engineering of disulfide bonds at the A2-A3 interface, despite having little impact on (predicted) domain architecture, did impart changes in FVIII-FIXa affinity and FX activation kinetics [64] suggesting this area is indeed important for the structure/function of FVIII. Early work [26] also highlighted a decrease in FVIII cofactor activity (FVIII:C) as well as a dissociation from its carrier protein von Willebrand Factor (VWF), another redox-labile protein that increases FVIII half-life in the circulation by ~6-fold [65]. In contrast to this, infusion of *N*-acetylcysteine (NAC) was associated with a short-lived rise in measurable FVIII activity, although this increase in FVIII activity was only observed in patients with an adverse reaction to the NAC infusion and a resultant rise in VWF release from the endothelium [66, 67]. There is also contradictory evidence in the literature regarding the effect of reduction on FIXa and the tenase complex (FIXa/FXa), whose formation is catalysed by FVIII. A more recent study investigating allosteric disulfide bond regulation of FIXa showed there was no change in FVIII or FIXa disulfide bond status when complexed, but did show a decrease in the low inherent activity of the tenase (FIXa/FX) complex when thioredoxin or PDI were present [67], again suggesting that reduction of FVIII supresses FVIII activity.

Our data lends weight to the argument that reduction of FVIII results in a decrease in activity and provides a structure-function rationale for this observation, i.e. reduction of the Cys1899-Cys1903 disulfide bond. Our modelling and experimental work indicates reduction of Cys1899-Cys1903 changes the structure of FVIII structure in a way that decreases its activity. In particular, the modelling suggests the largest structural changes are in the A1, A2, and A3 domains and this was experimentally validated with domain-specific antibodies. As FX and FIX bind these regions (C1 and C2 are predominantly for membrane association) it logically follows that the FVIII-FIX and FVIII-FX interactions are impacted by reduction-induced structural (allosteric) changes in FVIII. We were surprised to find that reduction of FVIII products resulted in both decreases *and* increases in activity measurements depending on which product was being evaluated. These observations may explain why there are contradictory reports in the literature about how FVIII reduction affects its activity. It is not clear at this stage what underpins these observed differences, but the FVIII products are all subtly different —different cellular expression systems (Chinese hamster ovary cells versus baby hamster kidney cells) and posttranslational modifications (glycosylation, sulfation, methionine oxidation, etc), different production and purification methods, and they are lyophilised and stored with

slightly different excipients—one product includes reduced glutathione for instance. Another possibility is that the structural changes in FVIII caused by the reduction of Cys1899-Cys1903 decreases FVIII activity but *increases* its association with phospholipids or its ability such that there is a decrease in FVIII-FIX complex formation but an overall increase in tenase complex formation. A similar effect has been shown in therapeutic monoclonal antibodies where reduction of labile dilsufides increases ligand binding but decreases Fc (fragment crystallisable) function, resulting in a decrease in drug efficacy. Further work will be needed to understand why different FVIII products behave differently, but it is a noteworthy observation that reduction/oxidation affects FVIII activity as conditions such as cancer and inflammation, in which thioredoxin levels are elevated, are associated with increased risk of thrombosis. As FVIII and many other haemostasis-related plasma proteins are thioredoxin substrates, it is possible that increased Trx activity may underpin the increased thrombosis risk in these pathophysiological conditions and further supports the development of oxidoreductase inhibitors to control thrombotic risk.

## Supporting information

**S1 Fig. Solvent accessibility of disulfide bonds during the MD simulations.** Boxplots showing the solvent accessible surface area (SASA) of the disulfide-forming cysteines during the MD simulations. The different panels refer to CD44 (A), tissue factor (B), CD132 (C), IL-4 (D), the D1 and D2 domains of GP130 (E), and FVIII light chains (F). In the boxplots, the top bar is the maximum observation, the lower bar is the minimum observation, the top of the box is the third quartile, the bottom of the box is the first quartile, while the middle bar represents the median. The medians are connected by a straight line to guide the eye.
(TIF)

**S2 Fig. Root mean square deviation of disulfide bonds during the MD simulations.** Boxplots showing the root mean square deviation (RMSD) of the disulfide bonds during the MD simulations. The different panels refer to CD44 (A), tissue factor (B), CD132 (C), IL-4 (D), the D1 and D2 domains of GP130 (E), and FVIII light chains (F). In the boxplots, the top bar is the maximum observation, the lower bar is the minimum observation, the top of the box is the third quartile, the bottom of the box is the first quartile, while the middle bar represents the median. The medians are connected by a straight line to guide the eye.
(TIF)

**S3 Fig. Labile disulfide bonds of CD44 and CD132 as predicted by DTT-CD44/CD132 interactions in MD simulations.** Histograms showing the probability that each of the (A) CD44 or (B) CD132 disulfide bonds is reduced by DTT according to the distance- (plain black bars) or distance+energy-based (dashed bars) criteria.
(TIF)

**S1 Table. Details of the peptides used in mass spectrometry quantitation of disulfide bond reduction.**
(XLSX)

**S1 Raw images.**
(PDF)

## Acknowledgments

Min Fang in the division of Analytical and Biological Sciences division at NIBSC for help with mass spectrometry. The authors wish to thank the hpc@polito team (http://www.hpc.polito.it),

and CINECA under the ISCRA initiative (Cys-Surf- HP10CSOLZQ), for the availability of high-performance computing and support.

## Author Contributions

**Conceptualization:** Andrea Arsiccio, Clive Metcalfe, Carmen Coxon.

**Formal analysis:** Andrea Arsiccio, Clive Metcalfe, Carmen Coxon.

**Investigation:** Andrea Arsiccio, Clive Metcalfe, Carmen Coxon.

**Methodology:** Andrea Arsiccio, Clive Metcalfe, Carmen Coxon.

**Project administration:** Carmen Coxon.

**Resources:** Roberto Pisano, Sanj Raut.

**Writing – original draft:** Carmen Coxon.

**Writing – review & editing:** Andrea Arsiccio, Clive Metcalfe.

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
