## [Decision Letter · Decision Letter 0]

20 Nov 2020

PONE-D-20-34221

A proximity-based in silico approach to identify redox-labile disulfide bonds: the example of FVIII

PLOS ONE

Dear Dr. Coxon,

Thank you for submitting your manuscript to PLOS ONE. After careful consideration, we feel that it has merit but does not fully meet PLOS ONE’s publication criteria as it currently stands. Therefore, we invite you to submit a revised version of the manuscript that addresses the points raised during the review process.

Although the two reviewers appreciated the fact that this work employed multiple approaches, they found that it is still too preliminary as it stands. Furthermore, considering that one of the main objectives of this work is  the development of a molecular dynamics protocol, more validation procedures should be undertaken.

Please, see attachment with the comments of reviewer #2 in a more appropriated format.

We look forward to receiving your revised manuscript.

Kind regards,

Luis E. S. Netto, PhD

Academic Editor

PLOS ONE

Journal Requirements:

5. Please upload a copy of Supporting Information Table S1 which you refer to in your text on page 7.

Reviewers' comments:

Reviewer's Responses to Questions

**Comments to the Author**

1. Is the manuscript technically sound, and do the data support the conclusions?

Reviewer #1: Yes

Reviewer #2: Partly

2. Has the statistical analysis been performed appropriately and rigorously? 

Reviewer #1: Yes

Reviewer #2: N/A

3. Have the authors made all data underlying the findings in their manuscript fully available?

Reviewer #1: Yes

Reviewer #2: Yes

4. Is the manuscript presented in an intelligible fashion and written in standard English?

Reviewer #1: Yes

Reviewer #2: No

5. Review Comments to the Author

Reviewer #1: The work submitted by A. Arisiccio and coworkers titled “A proximity-based in silico approach to identify redox-labile disulfide bonds: the example of FVIII” inferred the lability of protein disulfide bonds against TCEP reducing agent and explored by using different computational and experimental approaches the effect of the reduction.

The authors have studied several proteins, among them FVIII protein. They have investigated by means of mass spectrometry, HPLC, thiol modifications the lability of the disulfide bonds of a shorter variant of FVIII which laks b domain. Also, the authors have studied the interaction/recognition between oxidized and reducing forms of the full-length variant of FVIII and specific antibodies. Their findings are interesting, and they combined structural experiments with functional ones, a fact that makes the work more attractive. However, I think that the work must be improve prior to its publication.

- Authors should define more properly what 3-replica multiple walker parallel bias metadynamics is (in particular for the readers); they should explain how the bias-potentials are applied and how the sampling of the conformational space is enhanced by it.

- Why the authors did not analyze the solvent accessible surface area of the disulfide bonds along the MDs? They should include this parameter and the analysis. Other important parameters that I suggest to take into account: contacts and RMSF values corresponding to the S atoms.

- Given that TCEP is highly charged molecule, the authors should discuss whether this approach is specific for this reducing agent. Would the authors obtain similar results by using DTT? The interaction sites or the “residence time” for TCEP might be completely different compared to the expected for a significantly lower charged molecule. In this regard, may be important to analyze the electrostatic surfaces of the proteins. Thus, the molecular probe is an important issue. They should discuss this point.

- Units for TCEP concentration used in these experiments should be corrected and/or clarified (1mM or 1�M???, see that this information is critical, there are some typos like 1uM or even 1um along the paper).

- In figure 2 the authors should indicate that FVIII actually is B domain deleted BDD FVIII.

- Under the section “Experimental validation of in silico predicted redox-labile Cys1899-Cys1903”, please provide the correct reference for the online tool used (the reference indicated corresponds to a structure).

- Figure 5 and 6 should include the location of the disulfide bond that is reduced under TCEP treatment.

- ELISA results are very interesting. The reduction of recombinant full-length human FVIII resulted in a decrease in binding of the A2- and A3-specific anti-FVIII the C1 domain-specific antibody NB33 but no change in binding of the C2-specific antibody was observed. This is very attractive result!! It should be included in the abstract and discussion sections. Also, the authors should propose a hypothesis regarding the effect of this particularly labile disulfide bond on the conformation/stability/ or motions of the rest of the protein. They should discuss the term “allosteric disulfide bond” in this context.

- I think that the authors should calculate the fluctuations (RMSF) values along the simulations, to have an idea regarding the possibility that reductions results in an increase in molecular motions.

- Finally, I firmly suggest the authors to include a conformational control experiment (circular dichroism may be adequate): full-oxidized vs partially reduced forms.

- Minor: there are some typos that should be corrected. Revise the paper.

Reviewer #2: In "A proximity-based in silico approach to identify redox-labile disulfide bonds: the example of FVIII", Arsiccio and coworkers developed a MD protocol to identify labile disulfide bonds (DBs) in a small set of proteins with known structure, and then extrapolate the method and experimentally validated it by studying the redox-mediated effects of the coagulation factor VIII (FVIII). The agreement observed between the simulations and experimental evidence for the FVIII particular case is remarkable and well supported. Although Im not an expert, the significance of the biological problem regarding the role of labile disulfide bonds in the coagulation process is indubitable. However, the work is centered on the development of the MD protocol, and the score used by the authors is not validated and/or described-discussed enough to justify its future use in a broader sense. So, I do not recommend the publication of this work in its actual state. General and specific points to asses for future versions are described below.

General points:

1)As stated before, my main concern regarding this research, is the validity and convenience of the developed protocol to answer the DBs lability question. The authors have picked a proximity-based criterium using a distribution related parameter somehow indicating the local concentration of a probe (TCEP) near the DBs, that allow them to order (not identify unambiguosly) the lability of different DBs within a protein, in a small set of selected cases. In this sense, different topics should be addresed:

- A simple visual inspection of the DBs selected to study (Fig 2), led to the impression that DBs accesibility to any molecule (including solvent molecules) is a very important property to determine a particular DB lability (in terms of chemical tendency to get reduced). This idea is not new at all, the authors even discussed it when comparing the SASAs of the FVIII DBs (Table 2), please see these references in depth: https://doi.org/10.1016/j.bpj.2014.06.025;
https://doi.org/10.1098/rsos.171058. Furthermore, a simple calculation of the DBs SASA’s from the X-ray structures used by the authors suggest the same trend presented by the authors using the MD protocol:

protein pdb disulfide SASA(A2)

CD44 1uuh (chain A) 77-197 53,4

CD44 1uuh (chain A) 53-118 1,0

CD44 1uuh (chain A) 28-119 40,2

TF 1boy 186-209 36,1

TF 1boy 49-57 33,0

CD132 2erj (chain C) 160-209 18,1

CD132 2erj (chain C) 40-50 10,9

CD132 2erj (chain C) 80-93 0,7

IL-4 3bpl (chain A) 3-127 101,7

IL-4 3bpl (chain A) 46-99 12,6

IL-4 3bpl (chain A) 24-65 8,4

GP130 1p9m (chain A) 6-32 46,2

GP130 1p9m (chain A) 26-81 2,7

GP130 1p9m (chain A) 112-122 8

GP130 1p9m (chain A) 150-160 13,4

So, at this point I do not see the need of such complex metadynamics MD simulations to get an indicator of DBs lability. The authors need to justify much more deeply the use of the simulation protocol, identifying the advantages of such a use.

- It is my opinion that the validation of the MD protocol for it use in any DBs containing protein should include more DBs/protein examples, more protein folding types and DBs environments.

- The use of TCEP as the probe for the proximity criterium is based on its well known use as reductant. However, proximity probabilities arising from the MD simulations does not necesarly be related with reactivity. Added to the fact that the protocol is highlighting most solvent accesible DBs, the question that arises is: would any probe serve for the proximity criterium? In other terms, what are TCEP properties that justify its use? This aspect should be better discussed.

- Related with the previous point, the authors made use only of TCEP not only as the probe for the MD protocol, but also as non-physiological reducing agent. As authors may know, the characteristics of different reductants result in different reduction yields and specificities. For example, this fact is very well discussed in 10.1006/abio.1999.4203 for a DTT/TCEP comparison. So, I suggest to include at least DTT in the mass spectrometry and activity assays, in order to avoid possible biases.

- If maintening TCEP proximity criterium, I would advice to show the distributions in other fashion, boxplots maybe, as histograms as showed in this version does not seem to be the better/simpler way to compare the results between different DBs. In fact, they are not shown with the same bin width criterium for every system

2) Regarding the compartive study of FVIII dynamics and conformational changes upon C1899-1903 DB reduction: it seems to me that 100 ns of conventional MD is not enough sampling to characterize this phenomena, mainly because of the initial structure bias present for the reduced system. Although I acknowledge that the size of the system presents a limitation to achieve more exhaustive sampling, there are some strategies that can overcome this issue at least partially. I would advise to characterize much better the dynamic properties of the two systems, maybe by the use of some replica of accelerated MDs followed by conventional MDs or any other enhanced sampling method that the authors are familiar with. The global properties presented does not seem to be statistically well sampled, and so local properties such as HBs in particular protein regions would also present a big sampling bias.

Specific points:

- Through the document, there are several issues regarding references and the use of some reference manager software, please take care of it.

- There are also abbreviations and acronysms not defined previously, such as FVIII in the Abstract, DSB in the text…

- In the Introduction, there are some issues with capital letter use

- The format in which DBs are referred through out the text and figures is ambigous (Cys#-Cys#, Cys # - Cys #, Cys #-#, #-# disulfide bond...), I would advise to define and use one formal only

- Figure 5B is very low quality. Also, it is not easy to understand the reason why Figures 5B and 5C are not highlighting exactly the same regions of the protein, this question can be extended to Figure 6A. It seems that all 3 figures (5B, 5C and 6A) are meant to show basically the same observed changes, so I would only use Figure 5C with a clear representation of the scale that is being used.

- The definition of the different FVIII products in Figure 8 is not easy to understand and also is not clarified in the Figure caption.

6. PLOS authors have the option to publish the peer review history of their article (what does this mean?). If published, this will include your full peer review and any attached files.

Reviewer #1: No

Reviewer #2: No

---

## [Author Response · Author response to Decision Letter 0]

29 Apr 2021

A proximity-based in silico approach to identify redox-labile disulfide bonds: the example of FVIII

Andrea Arsiccio, Clive Metcalfe, Roberto Pisano, Sanj Raut, and Carmen Coxon

The authors would like to thank the reviewers for their accurate and fruitful revision of the present manuscript. Here, the point-by-point reply to reviewers’ comments is given, while changes made into the manuscript have been highlighted as colored text.

REVIEWER #1

GENERAL COMMENTS: The work submitted by A. Arisiccio and coworkers titled “A proximity-based in silico approach to identify redox-labile disulfide bonds: the example of FVIII” inferred the lability of protein disulfide bonds against TCEP reducing agent and explored by using different computational and experimental approaches the effect of the reduction.

The authors have studied several proteins, among them FVIII protein. They have investigated by means of mass spectrometry, HPLC, thiol modifications the lability of the disulfide bonds of a shorter variant of FVIII which laks b domain. Also, the authors have studied the interaction/recognition between oxidized and reducing forms of the full-length variant of FVIII and specific antibodies. Their findings are interesting, and they combined structural experiments with functional ones, a fact that makes the work more attractive. However, I think that the work must be improve prior to its publication.

SPECIFIC COMMENTS

COMMENT #1: Authors should define more properly what 3-replica multiple walker parallel bias metadynamics is (in particular for the readers); they should explain how the bias-potentials are applied and how the sampling of the conformational space is enhanced by it.

Author reply #1: Following the suggestions of reviewer 2, a new computational approach that makes use of unbiased MD simulations has been added to the revised version of the manuscript. This approach makes it possible to distinguish more clearly the contributions of solvent exposure and energy gain in disulfide reduction by TCEP. An energetic reweighting is, in fact, also implicit in the metadynamics simulations, but cannot be easily extracted from the simulations. Metadynamics simulations are, however, still used in the revised version of the manuscript to identify a proximity-based criterion for redox liability. A comment on parallel bias metadynamics has therefore been added to the revised version, 

“Metadynamics(37)works by introducing a history-dependent bias potential V(si,t) that acts on selected degrees of freedom of the system si, generally referred to as collective variables (CVs),

where ωi is a deposition hill height and σi a Gaussian width.

Such bias potential pushes the simulated system out of local minima, promoting the exploration of a considerably larger fraction of the phase space compared to normal MD simulations. The computational cost associated with the deposition of the bias, however, increases exponentially with the number of CVs selected. Parallel bias metadynamics alleviates this issue by constructing multiple one-dimensional biases, each acting on a single CV in parallel. This makes it possible to include as many CVs as needed at a reasonable computational cost.”

COMMENT #2: Why the authors did not analyze the solvent accessible surface area of the disulfide bonds along the MDs? They should include this parameter and the analysis. Other important parameters that I suggest to take into account: contacts and RMSF values corresponding to the S atoms.

Author reply #2: We have computed the solvent accessible surface area and average RMSD of the disulfides, as obtained from our MD simulations (Figures S1 and S2). Some comments on this have been added to the revised version of the manuscript,

“The average solvent accessible surface area (SASA) and root mean square deviation (RMSD) of each disulfide-forming cysteine during the unbiased simulations are displayed in Figures S1 and S2.”

“It is important to note that the distance-based criterion is not only a measure of solvent exposure, but also takes into account the specific interactions between the probe (TCEP) and the protein surface. For instance, in our simulations of CD44 with TCEP, the solvent accessibility of Cys77-Cys97 was similar to that of Cys28-Cys129, or even lower, as shown in Figure S1A.”

“Finally, it is also interesting to note, looking at Figure S2, that there is no evident correlation between the lability of a disulfide bond and its RMSD during the MD trajectories.”

COMMENT #3: Given that TCEP is highly charged molecule, the authors should discuss whether this approach is specific for this reducing agent. Would the authors obtain similar results by using DTT? The interaction sites or the “residence time” for TCEP might be completely different compared to the expected for a significantly lower charged molecule. In this regard, may be important to analyze the electrostatic surfaces of the proteins. Thus, the molecular probe is an important issue. They should discuss this point.

Author reply #3: CD44 and CD132 have been simulated both with TCEP and DTT in the revised version of the manuscript. The results with DTT have been supplied as Supporting Information (Figure S3). A comment on the importance of the molecular probe has been added to the revised version of the manuscript,

“We also studied the interactions between dithiothreitol (DTT) and CD44 or CD132 (Figure S3). DTT is smaller than TCEP (the radius of gyration of DTT is ≈0.26 nm, while it is ≈0.37 nm for TCEP). For this reason, a smaller cutoff (0.6 nm, instead of 0.8 nm) was used to estimate proximity of the DTT molecules. We found that the distance-based criterion could not distinguish between Cys77-Cys97 and Cys28-Cys129 in CD44 (Figure S3A), while adding the energy-based requirement succeeded in doing so, identifying Cys77-Cys97 as redox labile with 60% probability. The difference between Figures S3A (CD44-DTT interaction) and 4A (CD44-TCEP interaction) indicates that the selected probe, DTT or TCEP, plays a role (as different probes preferentially interact with different patches on the protein surface), although the ultimate conclusions on redox lability are not affected, when the distance+energy-based criterion is used. Figure S3B further shows that both the distance and distance+energy-based criteria succeeded in identifying Cys160-Cys209 in CD132 as redox labile also in the case of DTT as molecular probe.”

COMMENT #4: Units for TCEP concentration used in these experiments should be corrected and/or clarified (1mM or 1�M???, see that this information is critical, there are some typos like 1uM or even 1um along the paper).

Author reply #4: Many thanks for highlighting this. TCEP was always used at a 1 millimolar (mM) concentration. We have tried to make sure this is consistent throughout.

COMMENT #5: In figure 2 the authors should indicate that FVIII actually is B domain deleted BDD FVIII.

Author reply #5: We thank the reviewer for pointing this out. This information has been specified in the Figure under the relevant gels.

COMMENT #6: Under the section “Experimental validation of in silico predicted redox-labile Cys1899-Cys1903”, please provide the correct reference for the online tool used (the reference indicated corresponds to a structure).

Author reply #6: Apologies. There was a formatting error in the citations – this has now been corrected.

COMMENT #7: Figure 5 and 6 should include the location of the disulfide bond that is reduced under TCEP treatment.

Author reply #7: The Figure that shows the structural difference between full-oxidized and reduced FVIII (Figure 6C in the revised manuscript) now shows also the location of the disulfide bond that is reduced.

COMMENT #8: ELISA results are very interesting. The reduction of recombinant full-length human FVIII resulted in a decrease in binding of the A2- and A3-specific anti-FVIII the C1 domain-specific antibody NB33 but no change in binding of the C2-specific antibody was observed. This is very attractive result!! It should be included in the abstract and discussion sections. Also, the authors should propose a hypothesis regarding the effect of this particularly labile disulfide bond on the conformation/stability or motions of the rest of the protein. They should discuss the term “allosteric disulfide bond” in this context.

Author reply #8: Thank you for you comment. We have added the following to the discussion to highlight this result:

“In particular, the modelling suggests the largest structural changes are in the A1, A2, and A3 domains and this was experimentally validated with domain-specific antibodies. As FX and FIX bind these regions (C1 and C2 are predominantly for membrane binding) it logically follows that the FVIII-FIX and FVIII-FX interactions are impacted by reduction-induced structural (allosteric) changes in FVIII.”

COMMENT #9: I think that the authors should calculate the fluctuations (RMSF) values along the simulations, to have an idea regarding the possibility that reductions results in an increase in molecular motions.

Author reply #9: The unbiased simulations of FVIII have been extended to 200 ns, and the RMSF has been computed as suggested (Figure 6B in the revised version of the manuscript). We found no huge difference between full-oxidized and reduced FVIII, suggesting that reduction of Cys1899-Cys1903 does not increase the molecular motions,

“The root mean square fluctuations (RMSF) of the protein backbone (N-C�-C) atoms during the last 100 ns of the trajectories were also computed (Figure 6B). No huge difference was observed between the full-oxidized and reduced FVIII, suggesting that reduction of Cys1899-Cys1903 does not result in a noticeable increase in molecular motions.”

COMMENT #10: Finally, I firmly suggest the authors to include a conformational control experiment (circular dichroism may be adequate): full-oxidized vs partially reduced forms.

Author reply #10: Unfortunately we were unable to carry out any work pertaining to above experiment for two reasons, the first being that we did not have sufficient material to carry out CD or similar approaches, the second being that access to the site during the pandemic has been highly restricted. NIBSC is an official medicines control laboratory and because we are heavily engaged in both COVID vaccine release work and seasonal flu vaccine work, access to equipment and laboratories has been locked down for those involved in ‘priority work’, which, alas, does not include research.

COMMENT #11: Minor: there are some typos that should be corrected. Revise the paper.

Author reply #11: We thank the reviewer for pointing this out. The paper has been revised.

REVIEWER #2

GENERAL COMMENTS: In "A proximity-based in silico approach to identify redox-labile disulfide bonds: the example of FVIII", Arsiccio and coworkers developed a MD protocol to identify labile disulfide bonds (DBs) in a small set of proteins with known structure, and then extrapolate the method and experimentally validated it by studying the redox-mediated effects of the coagulation factor VIII (FVIII). The agreement observed between the simulations and experimental evidence for the FVIII particular case is remarkable and well supported. Although Im not an expert, the significance of the biological problem regarding the role of labile disulfide bonds in the coagulation process is indubitable. However, the work is centered on the development of the MD protocol, and the score used by the authors is not validated and/or described-discussed enough to justify its future use in a broader sense. So, I do not recommend the publication of this work in its actual state. General and specific points to asses for future versions are described below.

ANSWER TO GENERAL COMMENTS: We appreciate the constructive comments of the reviewer. The simulation approach has been revised as suggested, and we hope that the new version meets the expectations of the reviewer. The newly proposed simulation approach makes it possible to distinguish more clearly the roles of solvent exposure and energy gain in disulfide reduction. Using our approach, that builds on our previous findings, we can now not only order, but identify unambiguously the labile disulfide bond.

SPECIFIC COMMENTS

COMMENT #1: As stated before, my main concern regarding this research, is the validity and convenience of the developed protocol to answer the DBs lability question. The authors have picked a proximity-based criterium using a distribution related parameter somehow indicating the local concentration of a probe (TCEP) near the DBs, that allow them to order (not identify unambiguosly) the lability of different DBs within a protein, in a small set of selected cases. In this sense, different topics should be addresed:

- A simple visual inspection of the DBs selected to study (Fig 2), led to the impression that DBs accesibility to any molecule (including solvent molecules) is a very important property to determine a particular DB lability (in terms of chemical tendency to get reduced). This idea is not new at all, the authors even discussed it when comparing the SASAs of the FVIII DBs (Table 2), please see these references in depth: https://doi.org/10.1016/j.bpj.;
https://doi.org/10.1098/rsos.. Furthermore, a simple calculation of the DBs SASA’s from the X-ray structures used by the authors suggest the same trend presented by the authors using the MD protocol:

protein pdb disulfide SASA(A2)

CD44 1uuh (chain A) 77-197 53,4

CD44 1uuh (chain A) 53-118 1,0

CD44 1uuh (chain A) 28-119 40,2

TF 1boy 186-209 36,1

TF 1boy 49-57 33,0

CD132 2erj (chain C) 160-209 18,1

CD132 2erj (chain C) 40-50 10,9

CD132 2erj (chain C) 80-93 0,7

IL-4 3bpl (chain A) 3-127 101,7

IL-4 3bpl (chain A) 46-99 12,6

IL-4 3bpl (chain A) 24-65 8,4

GP130 1p9m (chain A) 6-32 46,2

GP130 1p9m (chain A) 26-81 2,7

GP130 1p9m (chain A) 112-122 8

GP130 1p9m (chain A) 150-160 13,4

So, at this point I do not see the need of such complex metadynamics MD simulations to get an indicator of DBs lability. The authors need to justify much more deeply the use of the simulation protocol, identifying the advantages of such a use.

Author reply #1: We understand the concern of the reviewers. However, the solvent accessibility of a static structure is not always completely representative of the protein dynamics. Moreover, the specific interaction of a probe, like TCEP, with a protein surface depends not only on the solvent accessibility, but also on the probe’s interactions with specific residues on the protein surface. This information is taken into account in our simulations, while it cannot be recovered from a simple analysis of the protein crystal structure. Finally, parallel bias metadynamics simulations also add a bias potential to the system’s energy, and an energy-based reweighting of each configuration being sampled is therefore implicitly hidden in the simulation setup. However, following the suggestions of the reviewer, a new simulation approach has been proposed in the revised version of the manuscript. The proximity-based criterion previously identified (i.e., number of TCEP molecules within 0.8 nm of the disulfide bond) has been integrated into a new strategy,

“For each of these proteins, we started at least 20 independent trajectories. Each protein was simulated in presence of 100 TCEP molecules, at 310 K and 1 bar, and using a 1 fs timestep. Simulation boxes were cubic, with side length equal to 10.7 nm for tissue factor, 10.2 nm for CD132, 7.9 nm for IL-4 and CD44, 10.6 nm for GP130 and 14.4 nm for FVIII. CD44 and CD132 were also simulated in presence of dithiothreitol (DTT), again at a 100:1 DTT:protein ratio. DTT is another common reducing agent, often used to break disulfide bonds in proteins. The topology file for DTT was also downloaded from the ATB server. The aggregated simulation time for these unbiased simulations was about 2.29 μs.

During these unbiased trajectories, we tracked the position of the TCEP (or DTT) molecules. We then applied two different criteria to identify a labile DSB (48), (1) distance-based or (2) distance+energy-based. According to the first criterion, a disulfide bond was considered to be reduced as soon as a TCEP molecule got closer than 0.8 nm. This same proximity-based condition had to be met for criterion number 2, but in this case a further requirement had to be satisfied. Specifically, the simulation was stopped, the disulfide bond was reduced and the resulting system energy minimized using the steepest descent algorithm. The initial structure, before disulfide bond reduction, was also energy minimized. The potential energies of the two systems, before (Ein) and after (Efin) disulfide reduction, were compared and the disulfide deemed to be broken only if Efin<Ein.”

This new approach has the advantage of being simpler to implement than the metadynamics simulations previously proposed. Moreover, it makes it possible to clearly decouple the roles of solvent exposure and energy gain in disulfide reduction. These two contributions cannot be easily distinguished in biased trajectories (like the metadynamics ones).

The new approach has been used for the model proteins selected in this work, and the importance of both the energy and distance criteria have been addressed,

“The two criteria (distance or distance+energy) were applied to 5 experimentally validated redox-labile disulfide bonds including CD44 (Figure 4A), TF (Figure 4B), CD132 (Figure 4C), IL-4 (Figure 4D), and the D1 and D2 domains of GP130 (Figure 4E). The distance-only criterion (plain black bars) was enough to identify unambiguously the labile disulfides Cys77-Cys97 in CD44 (Figure 4A), Cys186-Cys209 in TF (Figure 4B), Cys160-Cys209 in CD132 (Figure 4C), Cys3-Cys127 in IL4 (Figure 4D), and Cys6-Cys32 in GP130 (Figure 4E). In this case, adding the energy-based requirement (dashed bars) did not modify the main conclusions. For instance, the distance-only criterion predicted that Cys77-Cys97 in CD44 would have 96.7% probability to be reduced, and this probability further increased to 100% when the distance+energy-based criterion was applied. It is important to note that the distance-based criterion is not only a measure of solvent exposure, but also takes into account the specific interactions between the probe (TCEP) and the protein surface. For instance, in our simulations of CD44 with TCEP, the solvent accessibility of Cys77-Cys97 was similar to that of Cys28-Cys129, or even lower, as shown in Figure S1A. However, TCEP molecules mostly remained closer to Cys77-Cys97, presumably because of specific interactions with this patch on the protein surface. Moreover, it is important to note that the solvent accessibility, as measured in MD simulations, is representative of protein dynamics, as such being more reliable than the solvent accessibility measured from static protein structures.”

COMMENT #2: It is my opinion that the validation of the MD protocol for it use in any DBs containing protein should include more DBs/protein examples, more protein folding types and DBs environments.

Author reply #2: The proteins selected for this work (IL-4, CD44, CD132, GP130, TF and FVIII) belong to different protein folding types, and contain different DBs environments. It is the authors’ opinion that the chosen set is representative of a wide number of protein structures, allowing a robust validation of the simulation approach proposed.

COMMENT #3: The use of TCEP as the probe for the proximity criterium is based on its well known use as reductant. However, proximity probabilities arising from the MD simulations does not necesarly be related with reactivity. Added to the fact that the protocol is highlighting most solvent accesible DBs, the question that arises is: would any probe serve for the proximity criterium? In other terms, what are TCEP properties that justify its use? This aspect should be better discussed.

Author reply #3: As previously discussed, the proposed approach does not only highlight solvent exposure, but also takes into account specific interactions of the probe with the protein surface. This is also evidenced by a newly added set of simulations of CD44 or CD132 in presence of DTT as molecular probe,

 “We also studied the interactions between dithiothreitol (DTT) and CD44 or CD132 (Figure S3). DTT is smaller than TCEP (the radius of gyration of DTT is ≈0.26 nm, while it is ≈0.37 nm for TCEP). For this reason, a smaller cutoff (0.6 nm, instead of 0.8 nm) was used to estimate proximity of the DTT molecules. We found that the distance-based criterion could not distinguish between Cys77-Cys97 and Cys28-Cys129 in CD44 (Figure S3A), while adding the energy-based requirement succeeded in doing so, identifying Cys77-Cys97 as redox labile with 60% probability. The difference between Figures S3A (CD44-DTT interaction) and 4A (CD44-TCEP interaction) indicates that the selected probe, DTT or TCEP, plays a role (as different probes preferentially interact with different patches on the protein surface), although the ultimate conclusions on redox lability are not affected, when the distance+energy-based criterion is used. Figure S3B further shows that both the distance and distance+energy-based criteria succeeded in identifying Cys160-Cys209 in CD132 as redox labile also in the case of DTT as molecular probe.”

COMMENT #4: Related with the previous point, the authors made use only of TCEP not only as the probe for the MD protocol, but also as non-physiological reducing agent. As authors may know, the characteristics of different reductants result in different reduction yields and specificities. For example, this fact is very well discussed in 10.1006/abio.1999.4203 for a DTT/TCEP comparison. So, I suggest toinclude at least DTT in the mass spectrometry and activity assays, in order to avoid possible biases.

Author reply #4: As mentioned above, access to the site has been locked down for vaccine work (flu and covid) and batch release - all non-essential work (which frustratingly include research) has been shut down and we have been unable to carry out physical experimental work. Although frustrating, Andrea has carried out MD simulations with DTT and shown that the distance+energy-based criterion is effective at predicting disulfide bond lability for both TCEP and DTT.

COMMENT #5: If maintaining TCEP proximity criterium, I would advise to show the distributions in other fashion, boxplots maybe, as histograms as showed in this version does not seem to be the better/simpler way to compare the results between different DBs. In fact, they are not shown with the same bin width criterium for every system.

Author reply #5: As recommended by the reviewer, the distributions obtained by MD simulations are now shown as boxplots (Fig. 3-4, and S1-S2). In the boxplots, the top bar is the maximum observation, the lower bar is the minimum observation, the top of the box is the third quartile, the bottom of the box is the first quartile, while the middle bar represents the median. The medians are connected by a straight line to guide the eye.

COMMENT #6: Regarding the comparative study of FVIII dynamics and conformational changes upon C1899-1903 DB reduction: it seems to me that 100 ns of conventional MD is not enough sampling to characterize this phenomena, mainly because of the initial structure bias present for the reduced system. Although I acknowledge that the size of the system presents a limitation to achieve more exhaustive sampling, there are some strategies that can overcome this issue at least partially. I would advise to characterize much better the dynamic properties of the two systems, maybe by the use of some replica of accelerated MDs followed by conventional MDs or any other enhanced sampling method that the authors are familiar with. The global properties presented does not seem to be statistically well sampled, and so local properties such as HBs in particular protein regions would also present a big sampling bias.

Author reply #6: Unfortunately, enhanced sampling MD methods, like replica exchange (REMD) and metadynamics, can hardly be used for such large systems, as the computational cost would become prohibitive. For instance, running parallel tempering REMD with replicas in a temperature range of 300-450 K (which is typically used) would certainly require more than 100 replicas for such large system (the number of replicas required increases with the number of atoms in the simulation box). Such large number of replicas cannot be simulated at a reasonable speed. We however extended the simulation time to 200 ns, and updated the results accordingly (Fig. 6 in the revised manuscript). The trends of radius of gyration, number of internal hydrogen bonds, solvent accessible surface area and root mean square deviation shown in Figure 6 demonstrate that most conformational changes occur in the first 60 ns of the simulations, and afterwards the two systems start fluctuating around average values. 

COMMENT #7: Through the document, there are several issues regarding references and the use of some reference manager software, please take care of it.

Author reply #7:

COMMENT #8: There are also abbreviations and acronyms not defined previously, such as FVIII in the Abstract, DSB in the text…

Author reply #8: We thank the reviewer for pointing this out. The paper has been revised, and all the abbreviations properly defined before their first appearance.

COMMENT #9: In the Introduction, there are some issues with capital letter use

Author reply #9: The Introduction has been revised as suggested, and the issue with capital letters has been solved.

COMMENT #10: The format in which DBs are referred throughout the text and figures is ambigous (Cys#-Cys#, Cys # - Cys #, Cys #-#, #-# disulfide bond...), I would advise to define and use one formal only

Author reply #10: We thank the reviewer for pointing this out. The same format Cys#-Cys# is now used throughout the whole manuscript.

COMMENT #11: Figure 5B is very low quality. Also, it is not easy to understand the reason why Figures 5B and 5C are not highlighting exactly the same regions of the protein, this question can be extended to Figure 6A. It seems that all 3 figures (5B, 5C and 6A) are meant to show basically the same observed changes, so I would only use Figure 5C with a clear representation of the scale that is being used.

Author reply #11: The structural differences between full-oxidized and reduced FVIII are now shown in one Figure only (6C in the revised version of the manuscript, as suggested by the reviewer). A scale bar has been added.

COMMENT #12: The definition of the different FVIII products in Figure 8 is not easy to understand and also is not clarified in the Figure caption.

Author reply #12: Thanks for this. I have added a clearer description of each reagent in the legend. We hope this helps make it easier to understand to figure.

---

## [Decision Letter · Decision Letter 1]

18 May 2021

PONE-D-20-34221R1

A proximity-based in silico approach to identify redox-labile disulfide bonds: the example of FVIII

PLOS ONE

Dear Dr. Coxon,

Thank you for submitting your manuscript to PLOS ONE. After careful consideration, we feel that  this new version was very much improved , with several issues corrected and/or clarified. Unfortunately, there are still some points that still need to be clarified. Specifically,  both reviewers suggested addition of some experimental data such CD  and/or MS and activity assays.

Therefore, your manuscript has merit but does not fully meet PLOS ONE’s publication criteria as it currently stands. Therefore, we invite you to submit a revised version of the manuscript, carefully addressing the points raised by both reviewers.

We look forward to receiving your revised manuscript.

Kind regards,

Luis E. S. Netto, PhD

Academic Editor

PLOS ONE

Reviewers' comments:

Reviewer's Responses to Questions

**Comments to the Author**

1. If the authors have adequately addressed your comments raised in a previous round of review and you feel that this manuscript is now acceptable for publication, you may indicate that here to bypass the “Comments to the Author” section, enter your conflict of interest statement in the “Confidential to Editor” section, and submit your "Accept" recommendation.

Reviewer #1: All comments have been addressed

Reviewer #2: All comments have been addressed

2. Is the manuscript technically sound, and do the data support the conclusions?

Reviewer #1: Yes

Reviewer #2: Partly

3. Has the statistical analysis been performed appropriately and rigorously? 

Reviewer #1: N/A

Reviewer #2: Yes

4. Have the authors made all data underlying the findings in their manuscript fully available?

Reviewer #1: Yes

Reviewer #2: Yes

5. Is the manuscript presented in an intelligible fashion and written in standard English?

Reviewer #1: Yes

Reviewer #2: Yes

6. Review Comments to the Author

Reviewer #1: The new version of "A proximity-based in silico approach to identify redox-labile disulfide bonds: the example of FVIII" has been significantly improved compared to the previous one. The authors have included new simulations and analyses. From my viewpoint, the paper is a good contribution to study proteins and disulfide bond lability.

Reviewer #2: The revised version of the work by Arsiccio and coworkers is very much improved and several issues previously highlighted were corrected nd/or clarified. Importantly, the computational methodology is now much more clear together with the relevance of the use of such a "proximity based" approach using TCEP as the probe. Nevertheless, some points are still not correctly assesed, so I would suggest to discuss and/or correct them in future versions.

Specific points:

1. Although I think it is an editorial responsability to evaluate the implicances regarding restrictions to perform new experiments in these COVID times, I do think that experimental controls such CD (as suggested by reviewer 1) and/or MS and activity assays regarding the use of DTT are important for the present work

2. I agree the SASA measurements of static structures is "not always completely representative of the protein dynamics", however, is still a very good initial approximation of the property that is been measured, as highlighted in the previous review. That's the reason behind the suggestion of measuring SASA from the MDs, which as showed now in Fig S1, correlates well (obviously not perfectly) with DBs lability.

3. Although the authors made an effort to enlarge the statistical significance of the changes observed after reducing Cys1899-Cys1903 in FVIII, I still found the conclusions a little far-stretched, and much better sampling of the systems is needed. The author's arguments against the use of REMD may be reasonable and depends on the available computational capabilities, but how about accelerated MD? 200 ns long MDs does seem to be enough to explain allosteric (long distance) conformational changes of DB reduction in such more than a thousand amino acids protein. I am no saying what is being observed is anyway wrong, my reasoning is that the dynamical behavior is extremely biased by the initial structure, and the same is valid for any arising property. Thus, I sincerely think that if this issue is important for this particular research piece, much more sampling effort needs to be performed.

Regarding this point, Figure 6 needs some work:

i. RMSF is not informative at all

ii. It is quite difficult to extract information from FIg 6C. Taking aside the fact that the dynamical properties are not well sampled, Q parameter is an statical descriptor, which is also very dependent of protein alignment. Maybe this ref would be useful (10.1371/journal.pone.0119264). Also, I would suggest to align domains separately for these kinds of analysis, as the large size of the system sometimes difficults adequate alignment.

iii. Through out the text and this figure, the terms "disulfide present" and "disulfide broken" does not seem to be chemically correct, please consider to use oxidized and reduced.

Minor points:

1. Coloring of FVIII protein in Figure 1 is different from the other systems

2. Please consider to change the term "normal MD" to "conventional MD"

3. Separating ticks in x-axis of Fig 4 and Fig S3 are somehow missleading...also 0 and 100 values are difficult to observe in those graphs, plaese consider reviewing them, maybe changing scales

7. PLOS authors have the option to publish the peer review history of their article (what does this mean?). If published, this will include your full peer review and any attached files.

Reviewer #1: No

Reviewer #2: No

---

## [Author Response · Author response to Decision Letter 1]

17 Nov 2021

A proximity-based in silico approach to identify redox-labile disulfide bonds: the example of FVIII

Andrea Arsiccio, Clive Metcalfe, Roberto Pisano, Sanj Raut, and Carmen Coxon

The authors would like to thank the reviewers for their accurate and fruitful revision of the present manuscript. Here, the point-by-point reply to reviewers’ comments is given, while changes made into the manuscript have been highlighted as colored text.

REVIEWER #1

GENERAL COMMENTS: The new version of "A proximity-based in silico approach to identify redox-labile disulfide bonds: the example of FVIII" has been significantly improved compared to the previous one. The authors have included new simulations and analyses. From my viewpoint, the paper is a good contribution to study proteins and disulfide bond lability.

ANSWER TO GENERAL COMMENTS: We thank the reviewer for the revision of the present manuscript, which allowed us to improve our work.

REVIEWER #2

GENERAL COMMENTS: The revised version of the work by Arsiccio and coworkers is very much improved and several issues previously highlighted were corrected and/or clarified. Importantly, the computational methodology is now much more clear together with the relevance of the use of such a "proximity based" approach using TCEP as the probe. Nevertheless, some points are still not correctly addressed, so I would suggest to discuss and/or correct them in future versions.

ANSWER TO GENERAL COMMENTS: We very much appreciate the constructive comments of the reviewer. We have tried to address the comments raised by the referee, and hope the new version meets criteria for publication.

SPECIFIC COMMENTS

COMMENT #1: Although I think it is an editorial responsibility to evaluate the implicances regarding restrictions to perform new experiments in these COVID times, I do think that experimental controls such CD (as suggested by reviewer 1) and/or MS and activity assays regarding the use of DTT are important for the present work

Author reply #1: 

COMMENT #2: I agree the SASA measurements of static structures is "not always completely representative of the protein dynamics", however, is still a very good initial approximation of the property that is been measured, as highlighted in the previous review. That's the reason behind the suggestion of measuring SASA from the MDs, which as shown now in Fig S1, correlates well (obviously not perfectly) with DBs lability.

Author reply #2: The reviewer is right, and we appreciated the suggestion of measuring SASA from the MD simulations. A comment on this has been added to the revised manuscript,

“The average SASA of the different disulfide bonds correlates fairly well with their lability (see Figure S1), and is therefore a good initial approximation.”

COMMENT #3: Although the authors made an effort to enlarge the statistical significance of the changes observed after reducing Cys1899-Cys1903 in FVIII, I still found the conclusions a little far-stretched, and much better sampling of the systems is needed. The author's arguments against the use of REMD may be reasonable and depends on the available computational capabilities, but how about accelerated MD? 200 ns long MDs does seem to be enough to explain allosteric (long distance) conformational changes of DB reduction in such more than a thousand amino acids protein. I am not saying what is being observed is way wrong, my reasoning is that the dynamical behaviour is extremely biased by the initial structure, and the same is valid for any arising property. Thus, I sincerely think that if this issue is important for this particular research piece, much more sampling effort needs to be performed.

Regarding this point, Figure 6 needs some work:

i. RMSF is not informative at all

ii. It is quite difficult to extract information from FIg 6C. Taking aside the fact that the dynamical properties are not well sampled, Q parameter is an statical descriptor, which is also very dependent of protein alignment. Maybe this ref would be useful (10.1371/journal.pone.0119264). Also, I would suggest to align domains separately for these kinds of analysis, as the large size of the system sometimes difficult adequate alignment.

iii. Through out the text and this figure, the terms "disulfide present" and "disulfide broken" does not seem to be chemically correct, please consider to use oxidized and reduced.

Author reply #3: We understand the concerns of the reviewer. To give an idea of the computational cost required for REMD simulations of FVIII, if we aimed at using temperatures in the range 300-500 K, with an exchange probability between 0.2 and 0.3 (which is generally desired), according to the protocol described in (Alexandra Patriksson and David van der Spoel, A temperature predictor for parallel tempering simulations Phys. Chem. Chem. Phys., 10 pp. 2073-2077 (2008) http://dx.doi.org/10.1039/b716554d) we would need approximately 350 replicas. For each replica, we would need at least 31341 core hours to run 100 ns, which means that more than 10 million core hours would be required for the whole REMD simulation. This is only a rough estimate, and the computational cost may actually increase, because FVIII would denature at high temperatures, meaning that we would need to increase our box size and, consequently, the number of replicas. Unfortunately, such a high computational cost makes REMD simulations impracticable. For what concerns other accelerated MD simulations, such as umbrella sampling or metadynamics, they would require the choice of one (or more) collective variables to enhance the transition. Unfortunately, the choice of such collective variables is not easy when structure equilibration is desired. Generally, collective variables are used with the objective to push the system away from minima (for instance forcing the folding/unfolding transition) while we here aim at sampling as much as possible the energy minimum. For this reason, we would prefer not to try with enhanced sampling methods based on the biasing of collective variables, as we may risk pushing the system even further from the desired minimum. However, the reviewer is right about a possible bias in the results due to the initial structure, and we added a comment to the manuscript to warn the readers that, due to the very large size of FVIII, enhanced sampling methods were impracticable and the results of the conventional MD simulations presented may suffer from imperfect sampling.

“It is important to note that enhanced sampling methods were impracticable for these simulations due to the very large size of FVIII, and we therefore conducted conventional MD runs. We extended the simulation time until the computed properties (radius of gyration, internal hydrogen bonding network, solvent accessible surface area and backbone RMSD compared to the crystal structure, as shown in Figure 6) converged to stable values. The results of the conventional MD simulations presented are therefore a good indication of what may happen immediately after reduction of the labile disulfide bond; however, because of the impossibility to use enhanced sampling techniques, the reader should be aware that they may be not completely representative of long-time shifts in the protein conformation.”

As for the other comments,

i. we have removed the RMSF from Figure 6

ii. we agree with the reviewer that the Q parameter is a statical descriptor, but we applied it to the most sampled conformations in the equilibrated trajectories, as extracted by means of the Daura algorithm (Ref. 51 in the text). Such conformations are representative of more than 80% of the equilibrated trajectories, making the considerations drawn from the alignment more informative. However, as suggested by the reviewer, we also separately aligned the different domains of FVIII (new Figure 6C in the revised manuscript).

iii. We thank the reviewer for this comment, we modified the notation throughout the manuscript.

MINOR POINTS:

1. Colouring of FVIII protein in Figure 1 is different from the other systems

2. Please consider to change the term "normal MD" to "conventional MD"

3. Separating ticks in x-axis of Fig 4 and Fig S3 are somehow misleading...also 0 and 100 values are difficult to observe in those graphs, please consider reviewing them, maybe changing scales

REPLY TO MINOR POINTS:

1. We thank the reviewer for pointing this out. The colouring of FVIII has been changed.

2. The term ‘normal MD’ has been changed to ‘conventional MD’ as suggested.

3. Figs. 4 and S3 have been modified as recommended.

---

## [Decision Letter · Decision Letter 2]

1 Dec 2021

PONE-D-20-34221R2A proximity-based in silico approach to identify redox-labile disulfide bonds: the example of FVIII

PLOS ONE

Dear Dr. Coxon,

Thank you for submitting the revised version of your manuscript to PLOS ONE.

The manuscript was considerably improved, but still the improvement of Figure 6 is required. Figure 6 is really important and needs to be improved as previously discussed.

Therefore, we invite you to submit a revised version of the manuscript after addressing this minor point.

We look forward to receiving your revised manuscript.

Kind regards,

Luis E. S. Netto, PhD

Academic Editor

PLOS ONE

Journal Requirements:

Reviewers' comments:

Reviewer's Responses to Questions

**Comments to the Author**

1. If the authors have adequately addressed your comments raised in a previous round of review and you feel that this manuscript is now acceptable for publication, you may indicate that here to bypass the “Comments to the Author” section, enter your conflict of interest statement in the “Confidential to Editor” section, and submit your "Accept" recommendation.

Reviewer #2: (No Response)

2. Is the manuscript technically sound, and do the data support the conclusions?

Reviewer #2: Yes

3. Has the statistical analysis been performed appropriately and rigorously? 

Reviewer #2: Yes

4. Have the authors made all data underlying the findings in their manuscript fully available?

Reviewer #2: Yes

5. Is the manuscript presented in an intelligible fashion and written in standard English?

Reviewer #2: Yes

6. Review Comments to the Author

Reviewer #2: This version of the manuscript is much improved, and my advice is to publish after taking care of Figure 6. The authors states that Figure 6 was modified regarding my previous concerns, but I am not able to see the new version of the Figure (maybe a mistake on manuscript version). I think Figure 6 is really important and needs to be improved as previously discussed. I don't have additional comments.

7. PLOS authors have the option to publish the peer review history of their article (what does this mean?). If published, this will include your full peer review and any attached files.

Reviewer #2: No

---

## [Author Response · Author response to Decision Letter 2]

17 Dec 2021

Please find included the revised Figure6 - I had accidentally uploaded a previous version. Apologies for this and I hope the revised figure is agreeable. Many thanks for your patience.

---

## [Decision Letter · Decision Letter 3]

27 Dec 2021

A proximity-based in silico approach to identify redox-labile disulfide bonds: the example of FVIII

PONE-D-20-34221R3

Dear Dr. Coxon,

We’re pleased to inform you that your manuscript has been judged scientifically suitable for publication and will be formally accepted for publication once it meets all outstanding technical requirements.

Kind regards,

Luis E. S. Netto, PhD

Academic Editor

PLOS ONE

Additional Editor Comments (optional):

Reviewers' comments:

Reviewer's Responses to Questions

**Comments to the Author**

1. If the authors have adequately addressed your comments raised in a previous round of review and you feel that this manuscript is now acceptable for publication, you may indicate that here to bypass the “Comments to the Author” section, enter your conflict of interest statement in the “Confidential to Editor” section, and submit your "Accept" recommendation.

Reviewer #2: All comments have been addressed

2. Is the manuscript technically sound, and do the data support the conclusions?

Reviewer #2: (No Response)

3. Has the statistical analysis been performed appropriately and rigorously? 

Reviewer #2: (No Response)

4. Have the authors made all data underlying the findings in their manuscript fully available?

Reviewer #2: (No Response)

5. Is the manuscript presented in an intelligible fashion and written in standard English?

Reviewer #2: (No Response)

6. Review Comments to the Author

Reviewer #2: (No Response)

7. PLOS authors have the option to publish the peer review history of their article (what does this mean?). If published, this will include your full peer review and any attached files.

Reviewer #2: No

---

## [Editor Report · Acceptance letter]

24 Jan 2022

PONE-D-20-34221R3 

A proximity-based *in silico* approach to identify redox-labile disulfide bonds: the example of FVIII 

Dear Dr. Coxon:

I'm pleased to inform you that your manuscript has been deemed suitable for publication in PLOS ONE. Congratulations! Your manuscript is now with our production department. 

Kind regards, 

on behalf of

Dr. Luis E. S. Netto 

Academic Editor

PLOS ONE